# REVISITING THE ROLE OF LANGUAGE PRIORS IN VISION-LANGUAGE MODELS

## ABSTRACT

Vision-language models (VLMs) are impactful in part because they can be applied to a variety of visual understanding tasks in a zero-shot fashion, without any fine-tuning. We study currently popular *generative VLMs* that are trained for next-word generation given the image. We explore their zero-shot performance on the illustrative task of image-text retrieval across 8 popular vision-language benchmarks. Our first observation is that they can be repurposed for discriminative tasks (such as image-text retrieval) by simply computing the match score of generating a particular text string given an image. We call this probabilistic score the *Visual Generative Pre-Training Score* (VisualGPTScore). While the VisualGPTScore produces near-perfect accuracy on some retrieval benchmarks, it produces poor accuracy on others. We analyze this behavior through a probabilistic lens, pointing out that some benchmarks inadvertently capture unnatural language distributions by creating adversarial but unlikely text captions. In fact, we demonstrate that even a "blind" language model that ignores any image evidence can sometimes outperform all prior art, reminiscent of similar challenges faced by the visual-question answering (VQA) community many years ago. We derive a probabilistic post-processing scheme that controls for the amount of linguistic bias in generative VLMs at test time without having to retrain or fine-tune the model. We show that the VisualGPTScore, when appropriately debiased, is a strong zero-shot baseline for vision-language understanding, oftentimes producing state-of-the-art accuracy.

## 1 INTRODUCTION

Vision-language models (VLMs) trained on web-scale datasets will likely serve as the foundation for next-generation visual understanding systems. One reason for their widespread adoption is their ability to be used in an "off-the-shelf" (OTS) or zero-shot manner, without fine-tuning on any target application of interest. We study their OTS use on the task of image-text retrieval (e.g., given an image, predict which of $K$ possible captions is true) across a suite of 8 popular benchmarks.

**Challenges.** While the performance of foundational VLMs is impressive, many open challenges remain. Recent analysis (Kamath et al., 2023; Yuksekgonul et al., 2022) points out that leading VLMs such as CLIP (Radford et al., 2021) may often degrade to "bag-of-words" that confuse captions such as `"the horse is eating the grass"` and `"the grass is eating the horse"`. This makes it difficult to use VLMs to capture *compositions* of objects, attributes, and their relations. But somewhat interestingly, large-scale language models (LLMs) trained for autoregressive next-token prediction (Brown et al., 2020) seem to be able to capture such distinctions, which we investigate below. A related but under-appreciated difficulty is that of *benchmarking* the performance of visio-linguistic reasoning. Perhaps the most well-known example in the community is that of the influential VQA benchmarks (Antol et al., 2015), which could be largely solved by exploiting linguistic biases in the dataset – concretely, questions about images could often be answered by "blind" language-only models that did not look at the image (Goyal et al., 2017). Notably, we find that such blind algorithms can still produce strong performance on many contemporary image-text retrieval benchmarks where VLMs may struggle.

**Generative models for discriminative tasks.** We tackle the above challenges by revisiting the role of language priors through a probabilistic lens. To allow for a probabilistic treatment, we focus on generative VLMs that take an image as input and stochastically generate text via next-token

Scenario 1           Scenario 2

$t_1$= a white duck spreads its wings while in the water
$t_2$= a white wings spreads its water while in the duck
$t_3$= a white duck the its wings while in water spreads
$t_4$= white a duck spreads its wings in while the water
$t_5$= while in the spreads its wings water a white duck

$t_1$= people are posing in a kitchen
$t_2$= people are cooking in a kitchen

Figure 1: **Two train-test shifts encountered in image-to-text retrieval tasks.** Scenario 1 constructs negative text captions by shuffling words in the true caption (as in ARO-Flickr), but this produces implausible text such as `white a duck spreads its wings in while the water`. Here, exploiting the language bias of the training set will help since it will downweight the match score for negative captions. In fact, a blind language-only model can easily identify the correct caption. Scenario 2 constructs alternative text captions that are curated to be plausible (as in SugarCrepe). Here, the language bias of the training set may hurt, since it will prefer to match common captions (that score well under the language prior) as shown on the right.

prediction (Li et al., 2022; 2023). We first demonstrate that such models can be easily repurposed for discriminative tasks (such as retrieval) by setting the match score for an image-text pair to be the probability that the VLM would generate that text from the given image. We call this probability score the Visual Generative Pre-Training Score, or VisualGPTScore. Computing the VisualGPTScore is even more efficient than next-token generation since given an image, all tokens from a candidate text string can be evaluated in parallel. Though conceptually straightforward, such an approach (to our knowledge) has not been proposed in the literature. In fact, the generative VLMs that we analyze train *separate* discriminative heads for matching/classifying image-text pairs (Li et al., 2022), but we find that their language generation head itself produces better scores for matching (since it appears to better capture compositions). Indeed, OTS VisualGPTScore by itself performs surprisingly well on many benchmarks, even producing near-perfect accuracy on ARO (Yuksekgonul et al., 2022). But it still struggles on other benchmarks such as Winoground (Thrush et al., 2022). We analyze this below.

**The role of language priors.** We analyze the discrepancy in performance across benchmarks from a probabilistic perspective. Our key insight is that many benchmark biases can be formalized as mismatching distributions over text between train and test data - $P_{train}(\text{text})$ versus $P_{test}(\text{text})$. We use a first-principles analysis to account for distribution shift by simply reweighting the VisualGPTScore with the Bayes factor $P_{test}(\text{text})/P_{train}(\text{text})$, a process we call *debiasing*. To compute the Bayes reweighting factor, we need access to both the train and test language prior. We compute $P_{train}(\text{text})$ from an OTS VLM with Monte-Carlo samples of $P_{train}(\text{text}|\text{image})$ computed on trainset or Guassian noise images. Because $P_{test}(\text{text})$ may require access to the test set, we explore simplifying assumptions that assume it is (a) identical to $P_{train}(\text{text})$, (b) uninformative/uniform, or (c) tunable from a held-out val set. Our analysis helps explain the strong performance of the VisualGPTScore on certain benchmarks and its poor performance on others. Furthermore, this analysis provides simple strategies for improving performance with debiasing. We finally show a theoretical connection between debiasing and mutual information, which can be seen as a method for removing the effect of marginal priors when computing joint probability scores.

**Empirical Analysis.** We present an exhaustive empirical analysis of the OTS VisualGPTScore (and its debiased variants) for open-sourced image-conditioned language models (Li et al., 2022; 2023) across 8 popular vision-language benchmarks. We first point out that VisualGPTScore by itself produces SOTA accuracy on certain benchmarks like ARO (Yuksekgonul et al., 2022) where its inherent language bias helps remove incorrect text caption candidates that are also unnatural (such as `''a white duck the its wings while in water"` as shown in Fig. 1). In fact, we show that blind baselines also do quite well on such benchmarks, since language-only models can easily identify such poor captions. However, such language biases do not work well on benchmarks where incorrect caption candidates are also realistic. Here, VisualGPTScore should be debiased so as not to naively prefer more common captions that score well under its language prior. When given access to a val set that reveals the amount of language bias in the benchmark, debiasing consistently improves performance on benchmarks such as Flickr30K (Young et al., 2014) and Winoground (Thrush et al., 2022). Interestingly, we find that debiasing can also improve accuracy

on the *train* set used to learn the generative VLM, indicating that such models learn biased estimates of the true conditional distribution $P_{train}(\text{text}|\text{image})$. We describe this further in our appendix.

## 2 RELATED WORKS

**Vision-language modelling.** State-of-the-art VLMs like CLIP (Radford et al., 2021) are pre-trained on web-scale image-text datasets (Schuhmann et al., 2021; 2022) using discriminative objectives including image-text contrastive (ITC) (Radford et al., 2021; Jia et al., 2021) and image-text matching (ITM) (Li et al., 2021; 2022) loss, typically formulated as $P(\text{match}|\text{image}, \text{text})$. These pre-trained models exhibit robust zero-shot and few-shot (Lin et al., 2023; Wortsman et al., 2022) performance on traditional discriminative tasks (Deng et al., 2009; Lin et al., 2014), often on par with fully-supervised models. More recently, image-conditioned language models like Flamingo (Alayrac et al., 2022) and BLIP (Li et al., 2022; 2023) incorporate generative objectives (Bengio et al., 2003) primarily for downstream tasks such as captioning (Agrawal et al., 2019) and VQA (Goyal et al., 2017).

**Visio-linguistic compositionality.** Benchmarks like ARO (Yuksekgonul et al., 2022), Crepe (Ma et al., 2022), Winoground (Thrush et al., 2022), EqBen (Wang et al., 2023), VL-CheckList (Zhao et al., 2022), and SugarCrepe (Hsieh et al., 2023) show that discriminative scores of VLMs, such as ITCScore and ITMScore, fail on their image-text retrieval tasks that assess compositional reasoning. Concurrently, advances on these tasks often involve fine-tuning discriminative VLMs with more data. One of the most popular approaches, NegCLIP (Yuksekgonul et al., 2022), augments CLIP using programmatically generated negatives from original texts. Extending this, subsequent studies propose more expensive and heavily-engineered solutions. SyViC (Cascante-Bonilla et al., 2023) fine-tunes VLMs on million-scale synthetic images to augment spatial, attributive, and relation understanding. SGVL (Herzig et al., 2023) and Structure-CLIP (Huang et al., 2023) sample negatives using costly scene graph annotations. MosaiCLIP (Singh et al., 2023) and SVLC (Doveh et al., 2022) use linguistic tools such as scene graph parsers and LLMs to design better negative captions. The most recent DAC (Doveh et al., 2023) leverages a combination of foundation models including BLIP2, ChatGPT, and SAM to rewrite and augment image captions.

**Generative pre-training and scoring.** Vision models trained with *discriminative* objectives often lack incentives to learn structure information (Brendel & Bethge, 2019; Tejankar et al., 2021). Similarly, early LLMs trained with *discriminative* approaches, such as BERT (Devlin et al., 2018) and RoBERTa (Liu et al., 2019), have also been criticized as bag-of-words models insensitive to word order (Bertolini et al., 2022; Hessel & Schofield, 2021; Papadimitriou et al., 2022; Sinha et al., 2021). Conversely, generative pre-trained LLMs (Radford et al., 2019) demonstrate exceptional compositional understanding while pre-trained solely with a next-token prediction (Bengio et al., 2003) loss. Furthermore, generative scores of LLMs (OpenAI, 2023; Chung et al., 2022; Zhang et al., 2022) have flexible usage in downstream tasks, such as text evaluation (Yuan et al., 2021; Fu et al., 2023) and reranking (Keskar et al., 2019).

## 3 THE ROLE OF LANGUAGE PRIORS

In this section, we present a simple probabilistic treatment for analyzing the role of language priors in image-conditioned language models (or generative VLMs). Motivated by their strong but inconsistent performance across a variety of image-text retrieval benchmarks, we analyze their behavior when there exists a mismatch between training and test distributions, deriving simple schemes for addressing the mismatch with reweighting. We conclude by exposing a connection to related work on mutual information.

**Computing $P(\mathbf{t}|\mathbf{i})$.** To begin our probabilistic treatment, we first show that image-conditioned language models (that probabilistically generate text based on an image) can be repurposed for computing a score between a given image $\mathbf{i}$ and text caption $\mathbf{t}$. The likelihood of a text sequence $\mathbf{t} = \{t_1, t_2, \cdots, t_m\}$ conditioned on image $\mathbf{i}$ is naturally factorized as an autoregressive product (Bengio et al., 2003):

$$P(\mathbf{t}|\mathbf{i}) = \prod_{k=1}^{m} P(t_k|t_{<k}, \mathbf{i}) \tag{1}$$

Image-conditioned language models return back $m$ softmax distributions corresponding to the $m$ terms in the above expression. Text generation requires *sequential* token-by-token prediction, since token $t_k$ must be generated before it can be used as an input to generate the softmax distribution over token $t_{k+1}$. Interestingly, given an image $\mathbf{i}$ and text sequence $\mathbf{t}$, the above probability can be computed in *parallel* because the entire sequence of tokens $\{t_k\}$ are already available as input. We provide a visual illustration in Figure 2-a.

**Train-test shifts.** Given the image-conditioned model of $P(\mathbf{t}|\mathbf{i})$ above, we now analyze its behavior when applied to test data distributions that differs from the trainset, denoted as $P_{test}$ versus $P_{train}$. Recall that any joint distribution over images and text can be factored into a product over a language prior and an image likelihood $P(\mathbf{t}, \mathbf{i}) = P(\mathbf{t})P(\mathbf{i}|\mathbf{t})$. Our analysis makes the strong assumption that the image likelihood $P(\mathbf{i}|\mathbf{t})$ is identical across the train and test data, but the language prior $P(\mathbf{t})$ may differ. Intuitively, this assumes that the visual appearance of entities (such as a `"white duck"`) remains consistent across the training and test data, but the frequency of those entities (as manifested in the set of captions $P(\mathbf{t})$) may vary. We can now derive $P_{test}(\mathbf{t}|\mathbf{i})$ via Bayes rule:

$$P_{test}(\mathbf{t}|\mathbf{i}) \propto P(\mathbf{i}|\mathbf{t})P_{test}(\mathbf{t}) \tag{2}$$

$$= P(\mathbf{i}|\mathbf{t})\frac{P_{train}(\mathbf{t})}{P_{train}(\mathbf{t})}P_{test}(\mathbf{t}) \tag{3}$$

$$\propto P_{train}(\mathbf{t}|\mathbf{i})\frac{P_{test}(\mathbf{t})}{P_{train}(\mathbf{t})} \tag{4}$$

The above shows that the generative pre-training score $P_{train}(\mathbf{t}|\mathbf{i})$ need simply be weighted by the *ratio* of the language priors in the testset versus trainset. Intuitively, if a particular text caption appears *more* often in the testset than the trainset, one should *increase* the score reported by the generative model. However, one often does not have access to the text distribution on the testset. For example, real-world deployments and benchmark protocols may not reveal this. In such cases, one can make two practical assumptions; either the language distribution on test is identical to train, or it is uninformative/uniform (see Figure 1):

$$\text{Scenario 1:} \quad P_{test}(\mathbf{t}) = P_{train}(\mathbf{t}) \qquad \Rightarrow \qquad \text{Optimal score is } P_{train}(\mathbf{t}|\mathbf{i}). \tag{5}$$

$$\text{Scenario 2:} \quad P_{test}(\mathbf{t}) \text{ is uniform.} \qquad \Rightarrow \qquad \text{Optimal score is } \frac{P_{train}(\mathbf{t}|\mathbf{i})}{P_{train}(\mathbf{t})}. \tag{6}$$

**Tunable $\alpha$.** In reality, a testset might be a mix of both scenarios. To model this, we consider a soft combination where the language prior on the testset is assumed to be a flattened version of the language prior on the trainset, for some temperature parameter $\alpha \in [0, 1]$:

$$P_{test}(\mathbf{t}) \propto P_{train}(\mathbf{t})^{1-\alpha} \quad \Rightarrow \text{Optimal score is } \frac{P_{train}(\mathbf{t}|\mathbf{i})}{P_{train}(\mathbf{t})^\alpha} \tag{7}$$

By setting $\alpha$ to 0 or 1, one can obtain the two scenarios described above. Some deployments (or benchmarks) may benefit from tuning $\alpha$ on a val set.

**Implications for retrieval benchmarks.** We speculate some benchmarks like ARO-Flickr (Yuksekgonul et al., 2022) are close to scenario 1 because they include negative captions that are *implausible*, such as "`a white duck the its wings while in water spreads`". Such captions will have a low score under the language prior $P_{train}(\mathbf{t})$ and so reporting the raw generative score $P_{train}(\mathbf{t}|\mathbf{i})$ (that keeps its language prior or bias) will improve accuracy. In fact, we show that applying a *blind* language model (that ignores all image evidence) can itself often identify the correct caption. On the other hand, for test datasets with more *realistic* negative captions (scenario 2), it may be useful to remove the language bias of the trainset, since that will prefer to match to common captions (even if they do not necessarily agree with the input image). This appears to be the case for SugarCrepe (Hsieh et al., 2023), which uses LLMs like ChatGPT to ensure that the negative captions are realistic.

**Relationship to prior approaches.** Our approach to debiasing is reminiscent of mutual information, which can also be seen as a method for removing the effect of marginal priors when computing joint probability scores. In fact, our Appendix A derives that $\alpha$-debiasing is equivalent to a form of pointwise mutual information (PMI) known as $PMI^k$ for $k = \frac{1}{\alpha}$.

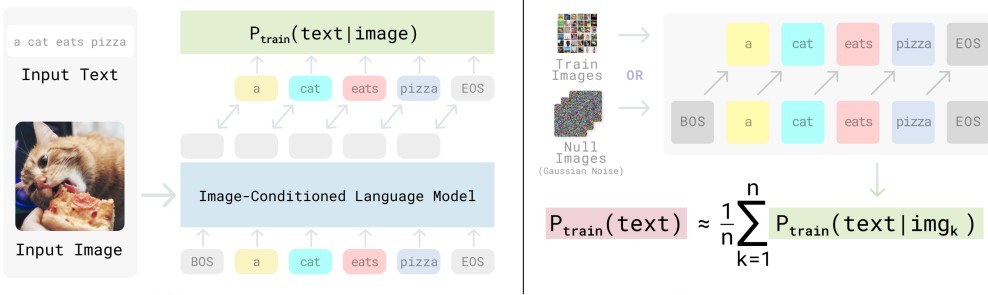

(a) $P_{train}(\mathbf{t}|\mathbf{i})$ through generative VLMs        (b) $P_{train}(\mathbf{t})$ via Monte Carlo sampling

Figure 2: **Estimating $P_{train}(\mathbf{t}|\mathbf{i})$ and $P_{train}(\mathbf{t})$ from generative VLMs.** Figure (a) shows how image-conditioned language models such as Li et al. (2022) that generate text based on an image can be repurposed for computing $P_{train}(\mathbf{t}|\mathbf{i})$, which is factorized as a product of $\prod_{k=1}^{m} P(t_k|t_{<k}, \mathbf{i})$ for a sequence of $m$ tokens. These terms can be efficiently computed in *parallel*, unlike *sequential* token-by-token prediction for text generation. Figure (b) shows two approaches for Monte Carlo sampling of $P_{train}(\mathbf{t})$. While the straightforward approach is to sample trainset images, we find that using as few as three "null" (Gaussian noise) images can achieve more robust estimates.

## 4 EXPERIMENTAL RESULTS ON I-TO-T RETRIEVAL

In this section, we verify our hypothesis on I-to-T retrieval benchmarks using state-of-the-art multimodal generative VLMs. In particular, we adopt image-conditioned language models such as BLIP (Li et al., 2022) as the learned estimator of $P_{train}(\mathbf{t}|\mathbf{i})$. Then, we discuss how we perform Monte Carlo estimation of $P_{train}(\mathbf{t})$, including a novel efficient sampling method based on "content-free" Gaussian noise images. Finally, we show the state-of-the-art results of our generative approach on existing I-to-T retrieval tasks.

**Preliminaries.** We leverage OTS image-conditioned language models (Yu et al., 2022; Alayrac et al., 2022; Li et al., 2023) to estimate $P_{train}(\mathbf{t})$. For ablation, we use the open-sourced BLIP models (Li et al., 2022), trained on public image-text corpora using discriminative (ITC and ITM) and generative (captioning) objectives. Discriminative objectives typically model $P(\text{match}|\mathbf{t}, \mathbf{i})$. For example, ITCScore calculates cosine similarity scores between image and text features using a dual-encoder; ITMScore jointly embeds image-text pairs via a fusion-encoder and returns softmax scores from a binary classifier. Lastly, we term the generative score as **Visual Generative Pre-Training Score** (**VisualGPTScore**). While BLIP is pre-trained using all three objectives, this generative score has not been applied to discriminative tasks before our work.

**Implementing VisualGPTScore.** Our method calculates an average of the log-likelihoods of $t_k$ at each token position $k$ and applies an exponent to cancel the log:

$$\textbf{VisualGPTScore}(\mathbf{t}, \mathbf{i}) := e^{\frac{1}{m} \sum_{k=1}^{m} \log(P(t_k|t_{<k}, \mathbf{i}))} \tag{8}$$

To condition on an input image, BLIP uses a multimodal casual self-attention mask (Li et al., 2022) in its image-grounded text decoder, i.e., each text token attends to all its preceding vision and text tokens. We emphasize that VisualGPTScore has the same computational cost as ITMScore, which uses the same underlying transformer but with a bi-directional self-attention mask to encode an image-text pair. We address potential biases of this estimator in Appendix C.

**Estimating $P_{train}(\mathbf{t})$ using Monte Carlo sampling (oracle approach).** Given $P_{train}(\mathbf{t}|\mathbf{i})$, we can estimate $P_{train}(\mathbf{t})$ via classic Monte Carlo sampling (Shapiro, 2003), by drawing $n$ images from the train distribution, such as LAION114M (Schuhmann et al., 2021) for BLIP:

$$P_{train}(\mathbf{t}) \approx \frac{1}{n} \sum_{k=1}^{n} P_{train}(\mathbf{t}|\mathbf{i}_k) \tag{9}$$

**Reducing sampling cost with content-free images (our approach).** The above Equation 9 requires many trainset samples to achieve robust estimates. To address this, we draw inspiration from (Zhao et al., 2021), which uses a *content-free* text prompt "N/A" to calibrate the probability of a text from LLMs, i.e., $P(\mathbf{t}|\text{"N/A"})$. To apply this to our generative VLMs, we choose to sample "null" inputs

| Score | Method | ARO | | | |
|---|---|---|---|---|---|
| | | Rel | Attr | COCO | Flickr |
| Random | - | 50.0 | 50.0 | 20.0 | 20.0 |
| Text-Only | Vera | 61.7 | 82.6 | 59.8 | 63.5 |
| | Grammar | 59.6 | 58.4 | 74.3 | 76.3 |
| $P_{LLM}(\mathbf{t})$ | BART | 81.1 | 73.6 | 95.0 | 95.2 |
| | Flan-T5 | 84.4 | 76.5 | 98.0 | 98.2 |
| | OPT | 84.7 | 79.8 | 97.9 | 98.6 |
| $P_{train}(\mathbf{t})$ | BLIP | 87.6 | 80.7 | 98.6 | 99.1 |
| $P(\text{match}\|\mathbf{t},\mathbf{i})$ | CLIP | 59.0 | 62.0 | 59.0 | 46.0 |
| | LAION2B-CLIP | 51.6 | 61.9 | 25.2 | 30.2 |
| | LAION5B-CLIP | 46.1 | 57.8 | 26.1 | 31.0 |
| | NegCLIP | 81.0 | 71.0 | 91.0 | 86.0 |
| | Structure-CLIP | 83.5 | 85.1 | - | - |
| | SyViC | 80.8 | 72.4 | 92.4 | 87.2 |
| | SGVL | - | - | 87.2 | 91.0 |
| | MosaiCLIP | 82.6 | 78.0 | 87.9 | 86.3 |
| | DAC-LLM | 81.3 | 73.9 | 94.5 | 95.7 |
| | DAC-SAM | 77.2 | 70.5 | 91.2 | 93.9 |
| | BLIP-ITC | 63.1 | 81.6 | 34.3 | 41.7 |
| | BLIP-ITM | 58.7 | 90.3 | 45.1 | 51.3 |
| $\frac{P_{train}(\mathbf{t}\|\mathbf{i})}{P_{train}(\mathbf{t})^{\alpha}}$ | Ours ($\alpha=0$) | 89.1 | 95.3 | 99.4 | 99.5 |
| | Ours ($\alpha=1$) | 68.1 | 87.9 | 32.4 | 44.5 |
| | Ours ($\alpha=\alpha^*$) | 89.1 | 95.4 | 99.4 | 99.5 |

**(a) Accuracy on ARO**

| Score | Method | VL-CheckList | | |
|---|---|---|---|---|
| | | Object | Attribute | Relation |
| Random | - | 50.0 | 50.0 | 50.0 |
| Text-Only | Vera | 82.5 | 74.0 | 85.7 |
| | Grammar | 58.0 | 52.4 | 68.5 |
| $P_{LLM}(\mathbf{t})$ | BART | 52.0 | 51.0 | 45.1 |
| | Flan-T5 | 60.3 | 55.0 | 49.3 |
| | OPT | 59.3 | 48.8 | 60.0 |
| $P_{train}(\mathbf{t})$ | BLIP | 68.2 | 58.7 | 75.9 |
| $P(\text{match}\|\mathbf{t},\mathbf{i})$ | CLIP | 81.6 | 67.6 | 63.1 |
| | LAION2B-CLIP | 84.7 | 67.8 | 66.5 |
| | LAION5B-CLIP | 87.9 | 70.3 | 63.9 |
| | NegCLIP | 81.4 | 72.2 | 63.5 |
| | SyViC | - | 70.4 | 69.4 |
| | SGVL | 85.2 | 78.2 | 80.4 |
| | SLVC | 85.0 | 72.0 | 69.0 |
| | DAC-LLM | 87.3 | 77.3 | 86.4 |
| | DAC-SAM | 88.5 | 75.8 | 89.8 |
| | BLIP-ITC | 90.6 | 80.3 | 73.5 |
| | BLIP-ITM | 89.9 | 80.7 | 67.7 |
| $\frac{P_{train}(\mathbf{t}\|\mathbf{i})}{P_{train}(\mathbf{t})^{\alpha}}$ | Ours ($\alpha=0$) | 92.6 | 78.7 | 90.8 |
| | Ours ($\alpha=1$) | 90.4 | 77.6 | 77.8 |
| | Ours ($\alpha=\alpha^*$) | 94.4 | 82.1 | 92.8 |

**(b) Accuracy on VL-CheckList**

| Score | Method | SugarCrepe | | |
|---|---|---|---|---|
| | | Replace | Swap | Add |
| Random | - | 50.0 | 50.0 | 50.0 |
| Text-Only | Vera | 49.5 | 49.3 | 49.5 |
| | Grammar | 50.0 | 50.0 | 50.0 |
| $P_{LLM}(\mathbf{t})$ | BART | 48.4 | 51.9 | 61.2 |
| | Flan-T5 | 51.4 | 57.6 | 40.9 |
| | OPT | 58.5 | 66.6 | 45.8 |
| $P_{train}(\mathbf{t})$ | BLIP | 75.9 | 77.1 | 70.9 |
| $P(\text{match}\|\mathbf{t},\mathbf{i})$ | CLIP | 80.8 | 63.3 | 75.1 |
| | LAION2B-CLIP | 86.5 | 68.6 | 88.4 |
| | LAION5B-CLIP | 85.0 | 68.0 | 89.6 |
| | NegCLIP | 88.3 | 76.2 | 90.2 |
| | BLIP-ITC | 85.8 | 73.8 | 85.7 |
| | BLIP-ITM | 88.7 | 81.3 | 87.6 |
| $\frac{P_{train}(\mathbf{t}\|\mathbf{i})}{P_{train}(\mathbf{t})^{\alpha}}$ | Ours ($\alpha=0$) | 93.3 | 91.0 | 91.0 |
| | Ours ($\alpha=1$) | 83.2 | 85.5 | 85.9 |
| | Ours ($\alpha=\alpha^*$) | 95.1 | 92.4 | 97.4 |

**(c) Accuracy on SugarCrepe**

| Score | Method | Crepe | | |
|---|---|---|---|---|
| | | Atom | Swap | Negate |
| Random | - | 16.7 | 16.7 | 16.7 |
| Text-Only | Vera | 43.7 | 70.8 | 66.2 |
| | Grammar | 18.2 | 50.9 | 9.8 |
| $P_{LLM}(\mathbf{t})$ | BART | 38.8 | 53.3 | 44.4 |
| | Flan-T5 | 43.0 | 69.5 | 13.6 |
| | OPT | 53.3 | 72.7 | 5.0 |
| $P_{train}(\mathbf{t})$ | BLIP | 55.4 | 69.7 | 60.8 |
| $P(\text{match}\|\mathbf{t},\mathbf{i})$ | CLIP | 22.3 | 26.6 | 28.8 |
| | LAION2B-CLIP | 23.6 | 24.8 | 18.0 |
| | LAION5B-CLIP | 24.2 | 23.9 | 20.1 |
| | BLIP-ITC | 24.8 | 17.7 | 26.5 |
| | BLIP-ITM | 29.5 | 20.7 | 25.5 |
| $\frac{P_{train}(\mathbf{t}\|\mathbf{i})}{P_{train}(\mathbf{t})^{\alpha}}$ | Ours ($\alpha=0$) | 73.2 | 78.1 | 79.6 |
| | Ours ($\alpha=1$) | 20.6 | 28.3 | 35.6 |
| | Ours ($\alpha=\alpha^*$) | 73.3 | 78.1 | 79.6 |

**(d) Accuracy on Crepe**

Table 1: **OTS generative VLMs are SOTA on image-to-text retrieval benchmarks.** We begin by evaluating blind language models (in red) . Surprisingly, this already produces SOTA accuracy on certain benchmarks such as ARO-Flickr, compared to the best discriminative approaches (in gray) . We also find that blind inference of generative VLMs, $P_{train}(\mathbf{t})$ via sampling Gaussian noise images (in blue) , often performs better and achieve above-chance performance even on the most recent SugarCrepe. Next, we show that simply repurposing a generative VLM's language generation head for computing image-text scores (VisualGPTScore in yellow) , which corresponds to $\alpha=0$, consistently produces SOTA accuracy across all benchmarks. Finally, debiasing this score by tuning $\alpha$ on val set (in green) further improves performance, establishing the new SOTA.

as Gaussian noise images. As a result, our approach requires as few as three images to compute Eq. 9 by sampling from Gaussian noise images with a mean of 0.4 and a standard deviation of 0.25. We find this method to be less computationally demanding and just as effective as sampling thousands of images from trainset. We provide a visual illustration of this method in Figure 2-b. We include sampling details in Appendix B.

**Benchmarks and evaluation protocols.** We comprehensively report on four popular I-to-T retrieval benchmarks, including ARO (Yuksekgonul et al., 2022), Crepe (Ma et al., 2022), SugarCrepe Hsieh et al. (2023), and VL-CheckList (Zhao et al., 2022). In these datasets, each image has a single positive caption and multiple negative captions. ARO (Yuksekgonul et al., 2022) has four datasets: VG-Relation, VG-Attribution, COCO-Order, and Flickr30k-Order. SugarCrepe (Hsieh et al., 2023) has three datasets: Replace, Swap, and Add. For Crepe (Ma et al., 2022), we use the entire productivity

set and report on three datasets: Atom, Negate, and Swap. VL-CheckList (Zhao et al., 2022) has three datasets: Object, Attribute, and Relation. We visualize all datasets in Appendix Table 13.

**SOTA performance on all four benchmarks.** In Table 1, we show that our OTS generative approaches, based on the BLIP model pre-trained on LAION-114M with ViT-L image encoder, achieves state-of-the-art results on all benchmarks. We outperform the best discriminative VLMs, including LAION5B-CLIP, and consistently surpass other heavily-engineered solutions, including NegCLIP, SyViC, MosaiCLIP, DAC, SVLC, SGVL, Structure-CLIP, all of which fine-tune CLIP on much more data. Details on how we report the baseline results can be found in Appendix E. For reference, we also include results of text-only Vera and Grammar from Hsieh et al. (2023). To show that even the most recent SugarCrepe is not exempt from language biases, we run two more text-only methods:

1. $P_{LLM}(\mathbf{t})$: passing captions into a pure LLM, such as BART-base (Yuan et al., 2021), FLAN-T5-XL (Chung et al., 2022), and OPT-2.7B (Zhang et al., 2022), to compute a text-only GPTScore (Fu et al., 2023).

2. $P_{train}(\mathbf{t})$: passing both captions and Gaussian noise images to BLIP as shown in Figure 2.

**Visualization of $\alpha$-tuning.** Finally, we observe that $\alpha$-tuning can consistently improve the performance. For visualization, we attach the results of $\alpha$-tuning in Table 2. We show side-by-side frequency charts of $P_{train}(\mathbf{t})$ for positive and negative captions.

## 5 ADDITIONAL EXPERIMENTAL RESULTS

In this section, we apply our OTS generative approaches to more benchmarks, including two compositionality benchmarks Winoground (Thrush et al., 2022) and EqBen (Wang et al., 2023), and two classic large-scale retrieval benchmarks COCO (Lin et al., 2014) and Flickr30K (Young et al., 2014). While naively applying VisualGPTScore leads to bad performance on these benchmarks, our training-free debiasing solution can consistently improve its performance with a held-out validation set. Furthermore, we derive the optimal text-to-image (T-to-I) retrieval objective and show that OTS generative scores can achieve robust T-to-I performance without debiasing.

**Evaluation protocols of Thrush et al. (2022).** While prior analysis (Diwan et al., 2022; Yuksekgonul et al., 2022) suggests that Winoground is too out-of-distribution to evaluate compositionality, we argue that evaluation protocols of Winoground and EqBen are more robust for future evaluations of VLMs. In these two benchmarks, each sample consists of two image-text pairs, ensuring **uniform image and text priors**. For simplicity, we consider a single Winoground sample: $(\mathbf{i}_0, \mathbf{t}_0)$ and $(\mathbf{i}_1, \mathbf{t}_1)$. The joint probabilities are $P_{test}(\mathbf{i}_0, \mathbf{t}_0) = P_{test}(\mathbf{i}_1, \mathbf{t}_1) = 0.5$. Meanwhile, $P_{test}(\mathbf{i}_0, \mathbf{t}_1) = P_{test}(\mathbf{i}_1, \mathbf{t}_0) = 0$. Applying the law of total probability gives $P_{test}(t_0) = P_{test}(t_1) = 0.5$. A similar derivation can show that image priors are uniform too. In addition, Winoground's evaluation metrics (text score and image score) penalize unimodal shortcut solutions. For example, in I-to-T retrieval, the *text score* gets 1 point only if *both images* are matched to the correct caption. Therefore, "blind" solutions that choose the same text regardless of images will get 0 text score. Similarly, for T-to-I retrieval, the *image score* gets 1 point only if *both captions* are matched to the correct image.

**Tuning $\alpha$ through cross validation.** In Table 3-a, we first show that OTS generative scores without debiasing ($\alpha$=0) lead to inferior performance on these I-to-T benchmarks. This confirms the importance of $\alpha$-tuning; even a simple $\alpha = 1$ can consistently and often significantly improve their I-to-T results. Furthermore, we try to use a held-out validation set to tune for optimal $\alpha \in [0, 1]$. We sample half of the data as validation set to search for $\alpha_{val}^*$ (using a step size of 0.001) and report the performance on the other half. We repeat this process 10 times to and report the mean and std. We observe that the optimal alpha is usually stable under the same dataset, regardless of the sampled val set. For COCO and Flickr30K, we perform $\alpha$-tuning using Recall@1 (R@1) on the official validation split. Because sampling additional Gaussian noise images can be too costly on these large-scale benchmarks, we directly approximate $P_{train}(\mathbf{t})$ by averaging the scores of testset images, without incurring any computational cost. More ablation studies such as $\alpha$-tuning using testset can be found in Appendix B. We also include the results of the ITMScore of BLIP for reference. While our debiasing solution can always boost performance, we observe that generative approaches still lag behind the ITMScore. This motivates us to study biases of generative scores towards more "common" texts in Appendix C.

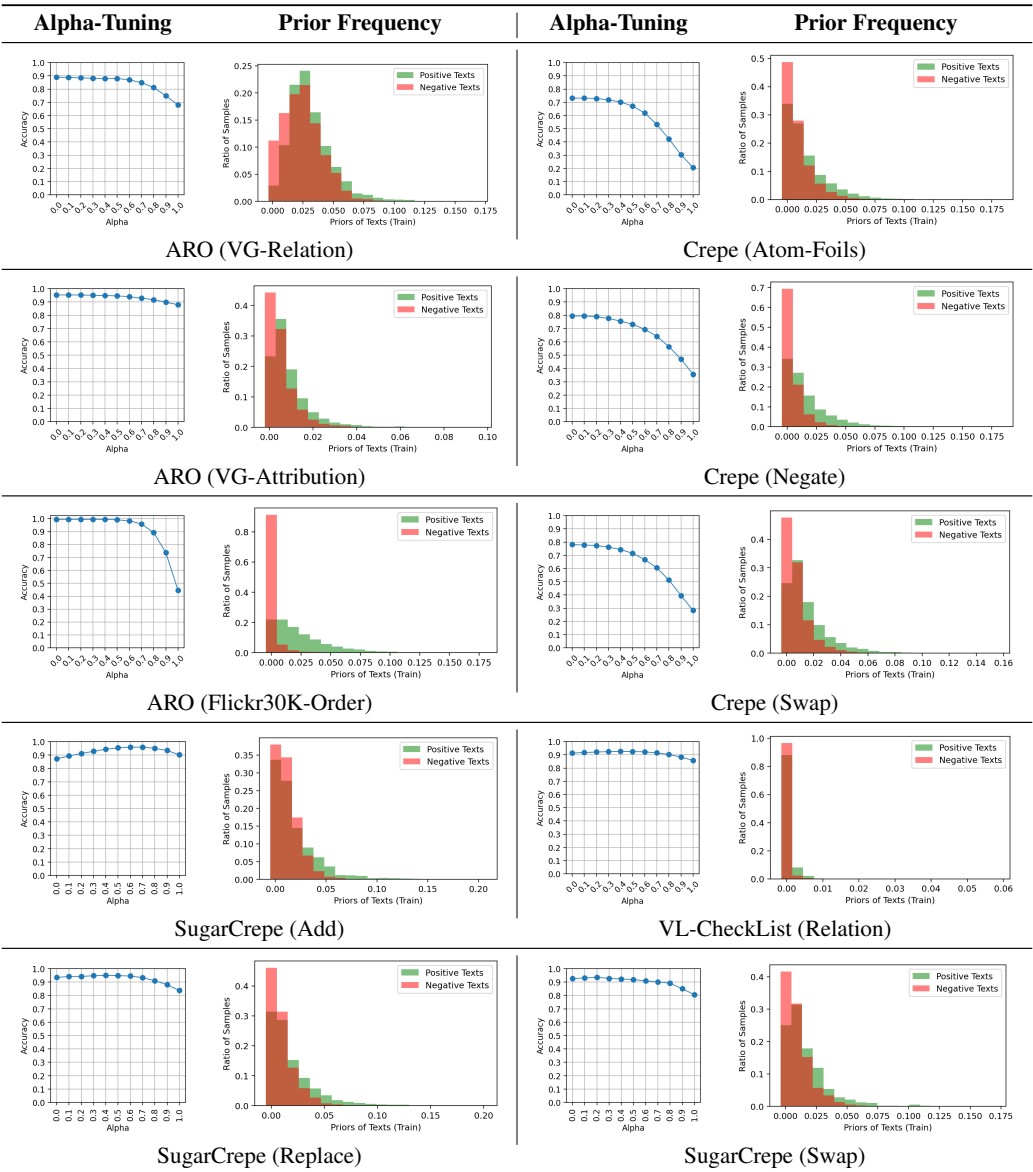

Table 2: $\alpha$-**tuning on I-to-T benchmarks and** $P_{train}(\mathbf{t})$ **frequency charts of both positive and negative captions.** Increasing $\alpha$ from 0 to 1 hurts performance on benchmarks with non-sensical negative captions such as ARO and Crepe. Such negative captions are easier to identify because of their low score under the language prior $P_{train}(\mathbf{t})$, implying such benchmarks may even be solved with blind algorithms that avoid looking at images. On the other hand, for benchmarks like SugarCrepe with more balanced $P_{train}(\mathbf{t})$ between positives and negatives, tuning $\alpha$ may lead to performance gain.

**Extending to T-to-I retrieval.** Though not the focus of our work, we also show that image-conditioned language models can be applied to T-to-I retrieval. Given a text caption $\mathbf{t}$, we can rewrite the Bayes optimal T-to-I retrieval objective as:

$$P_{test}(\mathbf{i}|\mathbf{t}) \propto P_{train}(\mathbf{t}|\mathbf{i}) * P_{train}(\mathbf{i}) \tag{10}$$

Equation 10 is hard to implement because we do not have access to $P_{train}(\mathbf{i})$. However, when $P_{train}(\mathbf{i})$ is approximately uniform, one can directly apply $P_{train}(\mathbf{t}|\mathbf{i})$ for optimal performance. We report T-to-I performance on all four benchmarks in Table 3-b, where our generative approach obtain competitive results compared against ITMScore, presumably because T-to-I retrieval is less affected by language biases.

| Metric | Benchmark | ITMScore | $\frac{P_{train}(\mathbf{t}|\mathbf{i})}{P_{train}(\mathbf{t})^\alpha}$ | | | |
|---|---|---|---|---|---|---|
| | | | $\alpha=0$ | $\alpha=1$ | $\alpha=\alpha^*_{val}$ | $\alpha^*_{val}$ |
| Text Score | Winoground | $35.5_{(2.4)}$ | $27.5_{(2.3)}$ | $33.7_{(2.4)}$ | $36.6_{(2.6)}$ | $0.855_{(0.023)}$ |
| | EqBen | $26.1_{(0.3)}$ | $9.6_{(0.2)}$ | $19.8_{(0.3)}$ | $19.8_{(0.3)}$ | $0.992_{(0.007)}$ |
| R@1/R@5 | COCO | 71.9 / 90.6 | 19.7 / 40.6 | 46.2 / 73.1 | 48.0 / 74.2 | 0.819 |
| | Flickr30k | 88.8 / 98.2 | 34.6 / 59.0 | 58.7 / 88.0 | 63.6 / 89.2 | 0.719 |

(a) $\alpha$-tuning on val sets for I-to-T retrieval

| Metric | Benchmark | ITMScore | $P_{train}(\mathbf{t}|\mathbf{i})$ |
|---|---|---|---|
| Image Score | Winoground | 15.8 | 21.5 |
| | EqBen | 20.3 | 26.1 |
| R@1/R@5 | COCO | 54.8 / 79.0 | 55.6 / 79.2 |
| | Flickr30k | 77.8 / 93.9 | 76.8 / 93.4 |

(b) T-to-I retrieval

Table 3: **Additional results on Winoground/EqBen/COCO/Flickr30K retrieval benchmarks.** Table (a) shows that tuning $\alpha$ can be essential for these compositionality and large-scale retrieval benchmarks. While OTS generative scores do not work well, debiasing with a larger $\alpha$ can consistently and often significantly improve I-to-T results on these tasks. To highlight the performance improvement, we mark results without debiasing ($\alpha = 0$) (in yellow), debiasing with a fixed $\alpha = 1$ (in pink), and cross-validation using held-out val sets ($\alpha = \alpha^*_{val}$) (in green). Table (b) shows that OTS generative scores can obtain favorable results on classic T-to-I retrieval tasks, competitive with the ITMScore.

## 6 DISCUSSION AND LIMITATIONS

**Summary.** Our study shows the efficacy of *generative* pre-training scores in solving *discriminative* tasks. With the rise of generative pre-training in recent models like GPT-4 (OpenAI, 2023), we see our work as a reliable starting point for future tasks. We present a first-principles analysis to account for mismatching distributions over text between train and test data. Based on this, we introduce a robust training-free (zero-shot) solution to debias linguistic priors in generative scores, achieving consistent and often significant improvement on all I-to-T retrieval tasks. Our thorough analysis also explains the performance discrepancy of generative scores on different benchmarks, and we hope it can encourage future work to revisit the issue of language biases in vision-language benchmarks.

**Limitations and future work.** Our approach depends on generative VLMs pre-trained on noisy web datasets, which may result in inherited biases (Mehrabi et al., 2021). We do not explore fine-tuning techniques due to computational constraints, but it is possible to improve the I-to-T retrieval performance using hard negative samples, such as with controllable generation (Keskar et al., 2019). Furthermore, our analysis is based on simplified assumptions. For instance, the image-conditioned language model might not accurately represent $P_{train}(\mathbf{t}|\mathbf{i})$, a phenomenon we examine in Appendix C. Estimating $P_{train}(\mathbf{t})$ by sampling Gaussian noise images can be suboptimal; future VLMs could directly model $P_{train}(\mathbf{t})$, or use techniques like coreset selection (Guo et al., 2022) or dataset distillation (Wu et al., 2023) to sample more representative images. Finally, we leave debiasing on the T-to-I retrieval task for future work.

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

# APPENDIX

## A    COMPARISON TO PMI$^k$

By assuming $P_{test}(\mathbf{t})$ to be a "flatten" version of $P_{train}(\mathbf{t})$, our Equation 7 can interpolate between scenario 1 (same train and test priors) and 2 (balanced test priors):

$$P_{test}(\mathbf{t}) \propto P_{train}(\mathbf{t})^{1-\alpha} \quad \Rightarrow \text{Optimal score is } \frac{P_{train}(\mathbf{t}|\mathbf{i})}{P_{train}(\mathbf{t})^{\alpha}} \tag{11}$$

In fact, the above equation can be rewritten using the language of PMI$^k$ (Role & Nadif, 2011; Daille, 1994), a well-known variant of PMI that controls the amount of debiasing (Li et al., 2016; Li & Jurafsky, 2016; Wang et al., 2020) in information retrieval:

$$\frac{P_{train}(\mathbf{t}|\mathbf{i})}{P_{train}(\mathbf{t})^{\alpha}} = \frac{P_{train}(\mathbf{t},\mathbf{i})}{P_{train}(\mathbf{i})P_{train}(\mathbf{t})^{\alpha}} \tag{12}$$

$$\propto \frac{P_{train}(\mathbf{t},\mathbf{i})^{\frac{1}{\alpha}}}{P_{train}(\mathbf{i})P_{train}(\mathbf{t})} \quad , \text{ as } P_{train}(\mathbf{i}) \text{ is constant in I-to-T} \tag{13}$$

$$= \text{pmi}^k_{P_{train}}(\mathbf{t},\mathbf{i}), \text{ where } k = \frac{1}{\alpha} \geq 1 \tag{14}$$

where

$$\text{pmi}_P(\mathbf{t},\mathbf{i}) = \frac{P(\mathbf{t},\mathbf{i})}{P(\mathbf{t})P(\mathbf{i})} = \frac{P(\mathbf{t}|\mathbf{i})}{P(\mathbf{t})} = \frac{P(\mathbf{i}|\mathbf{t})}{P(\mathbf{i})} \tag{15}$$

PMI is an information-theoretic measure that quantifies the *association* between two variables (Yao et al., 2010; Henning & Ewerth, 2017; Shrivastava et al., 2021). In the context of image-text retrieval, it measures how much more (or less) likely the image-text pair co-occurs than if the two were independent. Eq. 15 has found applications in diverse sequence-to-sequence modelling tasks (Wang et al., 2020; Li & Jurafsky, 2016; Li et al., 2016) as a retrieval (reranking) objective. Compared to the conditional likelihood $P(\mathbf{t}|\mathbf{i})$, PMI reduces the learned bias for preferring "common" texts with high marginal probabilities $P(\mathbf{t})$ (Li et al., 2016; Li & Jurafsky, 2016; Wang et al., 2020). This can be an alternative explanation for the effectiveness of our debiasing solutions.

## B    ABLATION STUDIES ON $\alpha$-TUNING

**Estimating $P_{train}(\mathbf{t})$ via null (Gaussian noise) images is more sample-efficient.**   We use Winoground to show that sampling Gaussian noise images to calculate $P_{train}(\mathbf{t})$ can be more efficient than sampling trainset images. As demonstrated in Table 4, a limited number of Gaussian noise images (e.g., 3 or 10) can surpass the results obtained with 1000 LAION images. Moreover, using null images produces less variance in the results.

| Sample Size | Guassian Noise Images | | Trainset Images | |
|---|---|---|---|---|
| | $\alpha=\alpha^*_{test}$ | $\alpha^*_{test}$ | $\alpha=\alpha^*_{test}$ | $\alpha^*_{test}$ |
| 3 | $35.95_{(0.5)}$ | $0.821_{(0.012)}$ | $32.20_{(1.6)}$ | $0.706_{(0.150)}$ |
| 10 | $36.25_{(0.4)}$ | $0.827_{(0.016)}$ | $33.60_{(0.9)}$ | $0.910_{(0.104)}$ |
| 100 | $36.35_{(0.1)}$ | $0.840_{(0.010)}$ | $34.70_{(0.6)}$ | $0.910_{(0.039)}$ |
| 1000 | $36.25_{(0.0)}$ | $0.850_{(0.000)}$ | $35.15_{(0.3)}$ | $0.960_{(0.033)}$ |

Table 4: **Comparing sampling of Gaussian noise images and trainset images for estimating $P_{train}(\mathbf{t})$.** We report text scores of $\alpha$-tuning on Winoground I-to-T retrieval task. We ablate 3/10/100/1000 Gaussian noise and LAION samples and report both mean and std using 5 sampling seeds. The optimal $\alpha^* \in [0, 1]$ is searched on testset via a step size of 0.001. The Gaussian noise images are sampled with a mean calculated from the LAION subset and a fixed std of 0.25.

**Details of Gaussian noise samples.**   Unless otherwise specified, the Gaussian noise images are sampled with a mean of 1.0 and a standard deviation of 0.25. By default, we use 100 images for Winoground, 30 images for EqBen, and 3 images for the rest of the benchmarks. We also fix the

sampling seed in our code to ensure reproducibility. We leave more advanced techniques of generating null images to future works.

**Alternative approach on COCO/Flickr30k: estimating $P_{train}(\mathbf{t})$ using testset images.** For large-scale retrieval benchmarks like COCO (Lin et al., 2014) and Flickr30k (Young et al., 2014), we can directly average scores of all candidate images (in the order of thousands) to efficiently approximate $P_{train}(\mathbf{t})$ without the need to sample additional images. This approach incurs zero computation cost as we have already pre-computed scores between each candidate image and text. We show in Table 5 that using testset images indeed results in better performance than sampling 3 Gaussian noise images.

| Metric | Benchmark | $P_{train}(\mathbf{t}\|\mathbf{i})$ | Sampling Method | $\frac{P_{train}(\mathbf{t}\|\mathbf{i})}{P_{train}(\mathbf{t})^\alpha}$ | | |
|---|---|---|---|---|---|---|
| | | | | $\alpha=1$ | $\alpha=\alpha^*_{val}$ | $\alpha^*_{val}$ |
| R@1 / R@5 | COCO | 19.7 / 40.6 | Testset Images | 46.2 / 73.1 | 48.0 / 74.2 | 0.819 |
| | | | Null Images | 24.4 / 52.6 | 40.4 / 66.6 | 0.600 |
| | Flickr30k | 34.6 / 59.0 | Testset Images | 58.7 / 88.0 | 63.6 / 89.2 | 0.719 |
| | | | Null Images | 27.8 / 62.2 | 48.5 / 79.0 | 0.427 |

Table 5: **I-to-T retrieval on COCO/Flickr30k using different sampling methods.** Estimating $P_{train}(\mathbf{t})$ by averaging the scores of testset images (with zero computational cost) demonstrates superior performance compared to sampling additional Gaussian noise images.

**Tuning $\alpha$ with a validation set.** In Table 6, similar performance trends are observed across validation and test splits of COCO and Flickr30k I-to-T retrieval benchmarks using the same $\alpha \in [0,1]$. Furthermore, $\alpha^*_{test}$ and $\alpha^*_{val}$ are empirically close. As such, our method can function as a reliable training-free debiasing method. Future studies may explore fine-tuning methods to further improve the debiasing performance.

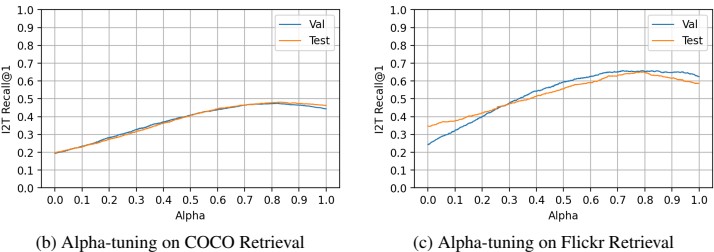

(b) Alpha-tuning on COCO Retrieval        (c) Alpha-tuning on Flickr Retrieval

Table 6: **$\alpha$-tuning results on both val set and test set for COCO/Flickr30k I-to-T retrieval.** We observe that validation and test performance are strongly correlated while we interpolate $\alpha \in [0,1]$.

## C  IS VISUALGPTSCORE A BIASED ESTIMATOR OF $P_{train}(\mathbf{t}|\mathbf{i})$?

**Retrieval performance on trainset (LAION).** This paper is built on the assumption that Visual-GPTScore is a reliable estimator of $P_{train}(\mathbf{t}|\mathbf{i})$. However, this simplifying assumption does not completely hold for the BLIP model we examine. We speculate that such OTS generative scores are biased towards more common texts. We witness this same phenomenon in Table 7, where we perform image-text retrieval on random subsets from training distribution LAION-114M (Li et al., 2022).

**Modelling the language bias in VisualGPTScore.** As evidenced in Table 7, we believe Visual-GPTScore is biased towards more common texts due to modelling error. To consider this error in our analysis, we rewrite the VisualGPTScore as:

$$\text{VisualGPTScore}(\mathbf{t},\mathbf{i}) := \hat{P}_{train}(\mathbf{t}|\mathbf{i}) = P_{train}(\mathbf{t}|\mathbf{i}) \cdot P_{train}(t)^\beta, \quad (16)$$

where $\hat{P}$ represents the (biased) model estimate and $P$ represents the true distribution. The model bias towards common texts is encoded by an unknown parameter $\beta$.

**Monte Carlo estimation using $\hat{P}$.** Because our Monte Carlo sampling method relies on $\hat{P}_{train}(\mathbf{t}|\mathbf{i})$, it is also a biased estimator of $P_{train}(\mathbf{t})$:

| Dataset Size | I-to-T Retrieval | | | | | T-to-I Retrieval | |
|---|---|---|---|---|---|---|---|
| | ITM | $\frac{P_{train}(\mathbf{t}\mid\mathbf{i})}{P_{train}(\mathbf{t})^\alpha}$ | | | | ITM | $P_{train}(\mathbf{t}\mid\mathbf{i})$ |
| | | $\alpha{=}0$ | $\alpha{=}1$ | $\alpha{=}\alpha^*$ | $\alpha^*$ | | |
| 100 | **96.0** | 59.0 | 94.0 | **95.0** | 0.535 | 95.0 | **97.0** |
| 1000 | **90.9** | 37.1 | 71.7 | 85.7 | 0.733 | 92.0 | **93.1** |
| 2000 | **87.2** | 32.8 | 62.3 | 64.3 | 0.840 | 87.8 | **89.8** |
| 5000 | **79.8** | 25.1 | 50.9 | 54.1 | 0.727 | 81.9 | **84.4** |

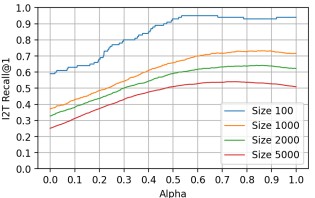

(a) Performance on LAION trainset retrieval    (b) Alpha-tuning on LAION

Table 7: **Retrieval performance on randomly sampled LAION114M subsets with varied sizes.** Table (a) shows that while OTS generative scores are robust for T-to-I retrieval, its performance degrades on I-to-T retrieval tasks when the number of candidate texts increases. This implies that OTS generative scores suffer from language biases towards certain texts even in the training set. Nonetheless, we show that our debiasing solution using either $\alpha = 1$ or optimal $\alpha^* \in [0, 1]$ with a step size of 0.001, can consistently boost the performance. Figure (b) visualizes $\alpha$-tuning results on LAION subsets, where each curve represents a different sample size.

$$\hat{P}_{train}(\mathbf{t}) := \frac{1}{n}\sum_{k=1}^{n}\hat{P}_{train}(\mathbf{t}\mid\mathbf{i}_k) = P_{train}(\mathbf{t})^{1+\beta}. \tag{17}$$

**Rewriting optimal I-to-T objective with $\hat{P}$.** We can rewrite Equation 4 as:

$$P_{test}(\mathbf{t}\mid\mathbf{i}) \propto P_{train}(\mathbf{t}\mid\mathbf{i})\frac{P_{test}(\mathbf{t})}{P_{train}(\mathbf{t})} \tag{18}$$

$$= \hat{P}_{train}(\mathbf{t}\mid\mathbf{i})\frac{P_{test}(\mathbf{t})}{P_{train}(\mathbf{t})^{1+\beta}} \tag{19}$$

$$= \hat{P}_{train}(\mathbf{t}\mid\mathbf{i})\frac{P_{test}(\mathbf{t})}{\hat{P}_{train}(\mathbf{t})} \tag{20}$$

$\alpha$-**tuning with $\hat{P}$.** Using Equation 20, we can reformulate $\alpha$-tuning (Equation 7) as follows:

$$P_{test}(\mathbf{t}) \propto P_{train}(\mathbf{t})^{1-\hat{\alpha}} \quad \Rightarrow \text{Optimal score is } \frac{\hat{P}_{train}(\mathbf{t}\mid\mathbf{i})}{\hat{P}_{train}(\mathbf{t})^\alpha} \tag{21}$$

where $\alpha = \frac{\hat{\alpha}+\beta}{1+\beta}$. Notably, the above equation has the same structure as before (Equation 7). This implies that even if $P_{train}(\mathbf{t}) = P_{test}(\mathbf{t})$, we still anticipate $\alpha = \frac{\beta}{1+\beta} \neq 0$. This accounts for why the optimal $\alpha$ is not 0 when we perform I-to-T retrieval on trainset in Table 7.

**Implication for vision-language modelling.** Our analysis indicates that similar to generative LLMs (Li et al., 2016; Li & Jurafsky, 2016), contemporary image-conditioned language models also experience issues related to imbalanced learning (Kang et al., 2019). Potential solutions could be: (a) refined sampling techniques for Monte Carlo estimation of $P(\mathbf{t})$ such as through dataset distillation (Wu et al., 2023), and (b) less biased modelling of $P(\mathbf{t}\mid\mathbf{i})$ such as through controllable generation (Keskar et al., 2019).

## D EXPERIMENTS WITH BLIP-2

We provide BLIP-2 results for completeness.

**BLIP-2 (Li et al., 2023) overview.** BLIP-2 leverages frozen pre-trained image encoders (Fang et al., 2022) and large language models (Chung et al., 2022; Zhang et al., 2022) to bootstrap vision-language pre-training. It proposes a lightweight Querying Transformer (Q-Former) that is trained in two stages. Similar to BLIP (Li et al., 2022), Q-Former is a mixture-of-expert model that can calculate ITC, ITM, and captioning loss given an image-text pair. Additionally, it introduces a set of trainable query tokens, whose outputs serve as *visual soft prompts* prepended as inputs to LLMs. In its first training stage, Q-Former is fine-tuned on the same LAION dataset using the same objectives

(ITC+ITM+captioning) as BLIP. In the second stage, the output query tokens from Q-Former are fed into a frozen language model, such as FLAN-T5 (Chung et al., 2022) or OPT (Chung et al., 2022), after a linear projection trained only with captioning loss. BLIP-2 achieves state-of-the-art performance on various vision-language tasks with significantly fewer trainable parameters.

**BLIP-2 results.** We present retrieval performance of the BLIP-2 model that uses ViT-L as the frozen image encoder. We report results for both the first-stage model (denoted as Q-Former) and the second-stage model which employs FLAN-T5 (Chung et al., 2022) as the frozen LLM.

| Benchmark | Dataset | Random | w. Q-Former | | | w. Flan-T5 |
|---|---|---|---|---|---|---|
| | | | ITC | ITM | $P_{train}(\mathbf{t}|\mathbf{i})$ | $P_{train}(\mathbf{t}|\mathbf{i})$ |
| ARO | VG-Relation | 50.0 | 46.4 | 67.2 | 90.7 | 89.1 |
| | VG-Attribution | 50.0 | 76.0 | 88.1 | 94.3 | 90.9 |
| | COCO-Order | 20.0 | 28.5 | 25.2 | 96.8 | 99.3 |
| | Flickr30K-Order | 20.0 | 25.3 | 28.6 | 97.5 | 99.7 |
| Crepe | Atom-Foils | 16.7 | 20.8 | 20.9 | 74.7 | 69.7 |
| | Negate | 16.7 | 13.4 | 14.2 | 79.1 | 90.0 |
| | Swap | 16.7 | 13.4 | 18.0 | 79.5 | 79.1 |
| VL-CheckList | Object | 50.0 | 89.7 | 89.2 | 90.1 | 84.1 |
| VL-CheckList | Attribute | 50.0 | 76.6 | 79.3 | 73.9 | 70.6 |
| VL-CheckList | Relation | 50.0 | 70.5 | 72.3 | 89.9 | 56.7 |
| SugarCrepe | Replace | 50.0 | 86.7 | 88.5 | 93.0 | 82.4 |
| SugarCrepe | Swap | 50.0 | 69.8 | 80.9 | 91.2 | 80.8 |
| SugarCrepe | Add | 50.0 | 86.5 | 88.0 | 92.7 | 76.2 |

Table 8: **BLIP-2 on ARO/Crepe/VL-CheckList/SugarCrepe.**

| Benchmark | Model | I-To-T (Text Score) | | | | | | T-To-I (Image Score) | | |
|---|---|---|---|---|---|---|---|---|---|---|
| | | ITC | ITM | $\frac{P_{train}(\mathbf{t}|\mathbf{i})}{P_{train}(\mathbf{t})^{\alpha}}$ | | | | ITC | ITM | $P_{train}(\mathbf{t}|\mathbf{i})$ |
| | | | | $\alpha{=}0$ | $\alpha{=}1$ | $\alpha{=}\alpha^*$ | $\alpha^*$ | | | |
| Winoground | BLIP | 28.0 | 35.8 | 27.0 | 33.0 | 36.5 | 0.836 | 9.0 | 15.8 | 21.5 |
| | BLIP2-QFormer | 30.0 | 42.5 | 24.3 | 29.3 | 33.0 | 0.882 | 10.5 | 19.0 | 20.0 |
| | BLIP2-FlanT5 | - | - | 25.3 | 31.5 | 34.3 | 0.764 | - | - | 19.5 |
| EqBen (Val) | BLIP | 20.9 | 26.0 | 9.6 | 19.8 | 19.8 | 0.982 | 20.3 | 20.3 | 26.1 |
| | BLIP2-QFormer | 32.1 | 36.2 | 12.2 | 21.9 | 22.2 | 0.969 | 23.4 | 28.4 | 26.6 |
| | BLIP2-FlanT5 | - | - | 8.5 | 22.0 | 22.0 | 1.000 | - | - | 20.9 |

Table 9: **BLIP-2 on Winoground/EqBen.**

# E  ADDITIONAL REPORTS

**Computational resources.** All experiments use a single NVIDIA GeForce 3090s GPU.

**Details of Table 1.** For CLIP, LAION2B-CLIP, and LAION5B-CLIP, we report the results from Hsieh et al. (2023) using the ViT-B-32, ViT-bigG-14, and xlm-roberta-large-ViT-H-14 models respectively. The results of NegCLIP, Structure-CLIP, SVLC, SGVL, DAC-LLM, and DAC-SAM are directly copied from their original papers. We run BLIP-ITC and BLIP-ITM using our own codebase, which will be released to the public.

**Group scores on Winoground/EqBen using BLIP (Table 10).**

| Method | Winoground | | | EqBen | | |
|---|---|---|---|---|---|---|
| | Text Score | Image Score | Group Score | Text Score | Image Score | Group Score |
| ITCScore | 28.0 | 9.0 | 6.5 | 20.9 | 20.3 | 10.6 |
| ITMScore | 35.8 | 15.8 | 13.3 | 26.0 | 20.3 | 12.6 |
| VisualGPTScore$^{\alpha^*}$ | 36.5 | 21.5 | 16.8 | 20.4 | 26.1 | 11.7 |

Table 10: Performance comparison of BLIP's ITCScore, ITMScore, and $\alpha$-tuned VisualGPTScore$^{\alpha^*}$ on Winoground (all) and EqBen (val).

**Fine-grained tags on Winoground (Table 11).**

**Performance on SugarCrepe (Table 12).**

| Dataset | Size | Method | Text Score | Image Score | Group Score |
|---|---|---|---|---|---|
| NoTag | 171 | ITCScore | 32.6 | 11.6 | 8.1 |
| | | ITMScore | 41.9 | 21.5 | 19.2 |
| | | VisualGPTScore$^{\alpha^*}$ | 43.0 | 28.5 | 23.8 |
| NonCompositional | 30 | ITCScore | 43.3 | 16.7 | 16.7 |
| | | ITMScore | 50.0 | 23.3 | 16.7 |
| | | VisualGPTScore$^{\alpha^*}$ | 43.3 | 33.3 | 26.7 |
| AmbiguouslyCorrect | 46 | ITCScore | 32.6 | 8.7 | 6.5 |
| | | ITMScore | 28.3 | 6.5 | 2.2 |
| | | VisualGPTScore$^{\alpha^*}$ | 26.1 | 19.6 | 8.7 |
| VisuallyDifficult | 38 | ITCScore | 29.0 | 7.9 | 7.9 |
| | | ITMScore | 26.3 | 10.5 | 7.9 |
| | | VisualGPTScore$^{\alpha^*}$ | 31.6 | 13.2 | 7.9 |
| UnusualImage | 56 | ITCScore | 32.5 | 8.9 | 8.9 |
| | | ITMScore | 21.4 | 10.7 | 7.1 |
| | | VisualGPTScore$^{\alpha^*}$ | 30.4 | 10.7 | 8.9 |
| UnusualText | 50 | ITCScore | 20.0 | 8.0 | 6.0 |
| | | ITMScore | 38.0 | 12.0 | 12.0 |
| | | VisualGPTScore$^{\alpha^*}$ | 30.0 | 18.0 | 12.0 |
| ComplexReasoning | 78 | ITCScore | 16.7 | 2.6 | 1.3 |
| | | ITMScore | 21.8 | 5.1 | 2.6 |
| | | VisualGPTScore$^{\alpha^*}$ | 21.8 | 10.3 | 6.4 |

Table 11: BLIP performance on Winoground subtags (Diwan et al., 2022). We report the number of test instances for each subtag and their respective text score, image score, group score.

| Method | Model | SugarCrepe | | | |
|---|---|---|---|---|---|
| | | **Replace** | **Swap** | **Add** | **AVG** |
| Human Performance | - | 98.67 | 99.50 | 99.00 | 99.06 |
| Random Chance | - | 50.00 | 50.00 | 50.00 | 50.00 |
| Text-Only Baseline | Vera | 49.46 | 49.30 | 49.50 | 49.42 |
| | Grammar | 50.00 | 50.00 | 50.00 | 50.00 |
| $P_{LLM}(\mathbf{t})$ | Bart | 48.41 | 51.93 | 61.16 | 53.83 |
| | Flan-T5 | 51.41 | 57.59 | 40.94 | 49.98 |
| | OPT | 58.53 | 66.58 | 45.78 | 56.96 |
| $P_{train}(\mathbf{t})$ | BLIP | 75.90 | 77.14 | 70.89 | 74.64 |
| ITCScore | CLIP-LAION2B | 86.50 | 68.56 | 88.37 | 81.14 |
| | CLIP-LAION5B | 84.98 | 67.95 | 89.62 | 80.85 |
| | BLIP | 85.76 | 73.79 | 85.66 | 81.74 |
| | BLIP-2 | 86.66 | 69.77 | 86.50 | 80.98 |
| | NegCLIP-SugarCrepe | 88.27 | 74.89 | 90.16 | 84.44 |
| ITMScore | BLIP | 88.68 | 81.29 | 87.57 | 85.85 |
| | BLIP2-Qformer | 88.45 | 80.87 | 87.96 | 85.76 |
| $P_{train}(\mathbf{t}|\mathbf{i})$ | BLIP | **93.33** | **91.00** | **90.98** | **91.77** |
| | BLIP2-Qformer | **93.00** | **91.24** | **92.69** | **92.31** |
| | BLIP2-FlanT5 | 82.44 | 76.57 | 76.24 | 78.42 |
| $\frac{P_{train}(\mathbf{t}|\mathbf{i})}{P_{train}(\mathbf{t})^{\alpha^*}}$ | BLIP | **95.09** | **92.39** | **97.36** | **94.95** |
| | BLIP2-Qformer | **94.62** | **92.27** | **97.58** | **94.82** |
| | BLIP2-FlanT5 | 85.69 | 78.80 | 91.76 | 85.42 |

Table 12: **Performance on SugarCrepe (Hsieh et al., 2023).** SugarCrepe is the most recent visio-linguistic compositionality benchmark which improves upon previous Crepe (Ma et al., 2022) by using state-of-the-art large language models (including ChatGPT), instead of rule-based templates, to generate more natural negative text captions. We show that text-only baselines and LLM-based methods indeed fail to succeed on SugarCrepe. However, our OTS generative approaches still achieve competitive results compared against SOTA discriminative approaches. The results of human performance, text-only baseline, and SOTA CLIP and NegCLIP-SugarCrepe are directly taken from the Hsieh et al. (2023). For other approaches, we evaluate their performance following the same procedure as described in main texts.

## F  BENCHMARK VISUALIZATION

We include random samples from each benchmark in Table 13.

| Dataset | Image | Positive Caption | Negative Caption(s) |
|---|---|---|---|
| VG-Relation | | the bus is to the right of the trees | the trees is to the right of the bus |
| VG-Attribution | | the striped zebra and the large tree | the large zebra and the striped tree |
| COCO-Order | | two dogs sharing a frisbee in their mouth in the snow | two frisby sharing a mouth in their snow in the dogs
in dogs the in frisby sharing two mouth their a snow
two dogs sharing in a frisby their mouth in snow the
a frisby in the snow two dogs sharing their mouth in |
| Flickr30K-Order | | a white duck spreads its wings while in the water | a white wings spreads its water while in the duck
a white duck its wings while in water spreads
white a duck spreads its wings in while the water
while in the spreads its wings water a white duck |
| SugarCrepe Add-Attribute | | They are going to serve pizza for lunch today. | They are going to serve pizza topped with pineapple for lunch today. |
| SugarCrepe Add-Object | | A man kisses the top of a woman's head. | A man kisses the top of a woman's head with a flower in his hand. |
| SugarCrepe Replace-Attribute | | A kid standing with a small suitcase on a street. | A kid standing with a big suitcase on a street. |
| SugarCrepe Replace-Object | | A duck floating in the water near a bunch of grass and rocks | A swan floating in the water near a bunch of grass and rocks. |
| SugarCrepe Replace-Relation | | A clock tower stands in front of a large mirrored sky scraper. | A clock tower stands behind a large mirrored sky scraper. |
| SugarCrepe Swap-Attribute | | A tennis player is taking a swing on a red court. | A red player is taking a swing on a tennis court. |
| SugarCrepe Swap-Object | | A woman holding a game controller with a man looking on. | A man holding a game controller with a woman looking on. |
| Crepe-AtomFoils | | microwave in a kitchen, and sink in a kitchen. | microwave in a cupboard, and sink in a kitchen
microwave in a bar, and sink in a kitchen
line in a kitchen, and sink in a kitchen
microwave in a kitchen, and shower in a kitchen
microwave in a kitchen, and tap in a kitchen |
| Crepe-Negate | | a chair next to a table, with the back of the chair visible. | A chair is not next to a table, with the back of the chair visible
A chair next to a table, with the back not of the chair visible
A chair next to a table, with the back of the chair visible
A chair next to a table, with something of the chair visible. There is no back.
There is no chair next to a table, with the back of the chair visible |
| Crepe-Swap | | a car driving on a road with a line next to a tree. | a car driving on a bright green leaves with a line next to a tree
a bright green leaves driving on a road with a line next to a tree
a car driving on a tree with a line next to a road
a car driving on a road with a line next to a white car
a car driving on a road with a line next to a street |
| VL-CheckList Relation (spatial) | | person read book | person carry book |
| VL-CheckList Relation (action) | | sign near boy | sign far from book |
| Winoground | | a person on top of the world | the world on top of a person |
| | | the world on top of a person | a person on top of the world |
| EqBen | | The person is touching the dish which is in front of him/her. | The person is holding the dish which is in front of him/her. |
| | | The person is holding the dish which is in front of him/her. | The person is touching the dish which is in front of him/her. |

Table 13: **Visualization of benchmarks.** ARO (VG-Relation/VG-Attribution/COCO-Order/Flickr30K-Order), Crepe (AtomFoils/Negate/Swap), VL-CheckList (Object/Attribute/Relation), SugarCrepe (Replace/Swap/Add) are constructed by generating hard negative captions for an image-text pair. On the other hand, each sample of Winoground and EqBen has two image-text pairs.

