# OpenReview forum: "Revisiting the Role of Language Priors in Vision-Language Models"
_ICLR.cc/2024/Conference — ICLR 2024 Conference Desk Rejected Submission_

### Official Review · Reviewer_AoET · 2023-10-30

**Soundness:** 2 fair
**Presentation:** 3 good
**Contribution:** 3 good
**Rating:** 5
**Confidence:** 3

**Summary:**

This paper focuses on the generative vision-language models, which have been the focus of many recent works. The authors introduce a novel approach, VisualGPTScore, for employing these generative models in discriminative tasks, particularly image-text alignment and retrieval. Empirical experiments suggest that blind language models occasionally outperform established methodologies. Building on this insight, the authors propose an additional post-processing step during testing to control "language bias." In essence, this paper presents a promising avenue for harnessing generative model confidences effectively.

**Strengths:**

* The paper introduces a probabilistic approach to generative model prediction confidences, exhibiting superior performance in comparison to Image Text Matching (ITM) formulations.
* Rigorous experimentation and ablation analyses showcase the efficacy of VisualGPTScore.
* The paper maintains a well-structured and articulate presentation.

**Weaknesses:**

* The paper's results under standard training and testing assumptions ($\alpha = 0$) are exceptional, but the rationale for deviating from this assumption lacks proper motivation. Measuring $\alpha^*$ demands test-time privileged information, which raises concerns about the approach's validity.

* Addressing language bias is a crucial aspect, yet the method employed for debiasing, measured by $\alpha$, appears to operate at the dataset level ($P_{test}$ vs. $P_{train}$) rather than the instance level. Evaluating the total effect, as $P(t|i) - P(t|i=\phi)$, would provide a more meaningful approach to remove the “language bias”.

* While the analysis is extensive in the context of I-to-T retrieval tasks, it falls short in terms of assessing a broader range of downstream tasks. Incorporating analyses of zero-shot classification, VQA2.0, and GQA tasks would offer a more comprehensive perspective.

**Questions:**

* If privileged information is employed to measure $\alpha^*$, how does this impact individual biases such as "black apple" vs "red apple"? Shouldn't the value of  $\alpha^*$ vary based on the specific target ($t$)?
* What is the performance of VisualGPTScore on zero-shot classification? This would contribute to a more comprehensive evaluation of the proposed approach.
* In the case of "blind models," how is the measurement of $P(t^i_{positive})$ with respect to $P(t^i_{negative})$ conducted for each $i^{th}$ instance? Elaboration on the methodology for evaluating individual test instances in “blind models” is needed.
* Is it plausible that negative captions rarely occur in web corpora? This factor might be affecting the performance of "blind models" and deserves further investigation.

---

> ### Author Response · Authors · 2023-11-14
> **Rebuttal**
>
> We appreciate Reviewer AoET's feedback. We begin by clearing up some potential misunderstandings:
>
> > **Question 3: In the case of "blind models", how is the measurement of $P(t^i_{positive})$ with respect to $P(t^i_{negative})$ conducted for each instance? Elaboration on the methodology for evaluating individual test instances in "blind models" is needed.**
>
> $P_{train}(text)$ is measured for each individual text independently using Monte Carlo estimation (Equation 9): $P_{train}(text) \approx \frac{1}{n} \sum_{k=1}^n P_{train}(text | image_k)$, where $k$ represents the number of random Gaussian noise images sampled.
>
> > **Weaknesses 1: Measuring $\alpha^{*}$ demands test-time privileged information, which raises concerns about the approach's validity.**
>
> Our method **does not demand test-time privileged information**. In practice, one only needs a small validation set from the target task to search for optimal alpha, or can simply make practical assumptions without tuning alpha, e.g., choosing $\alpha=1$. We show in Table 2-a that both consistently improve performance without using test-time privileged information.
>
> > **Weaknesses 2 + Question 1: Addressing language bias is a crucial aspect, yet the method employed for debiasing, measured by a, appears to operate at the dataset level ($P_{train}$ vs. $P_{test}$) rather than the instance level. Evaluating the total effect, as P(t|i) - P(t|i =$\phi$), would provide a more meaningful approach to remove the "language bias". Shouldn't the value of $a^{*}$ vary based on the specific target (t)?**
>
> Our $\alpha-$debiasing method ($\frac{P_{train}(text | image)}{P_{train}(text)^\alpha}$) is motivated by the widely existing **train-test shift** from VLM’s training set (web corpora) to testing benchmarks (such as SugarCrepe), as thoroughly discussed in Section 3. Without additional training, it is unclear how to measure an $\alpha$ that varies based on specific target texts. As our paper focuses on training-free debiasing, we leave learning instance-specific $\alpha$ to future work.
>
> > **Weaknesses 3 + Question 2: While the analysis is extensive in the context of I-to-T retrieval tasks, it falls short in terms of assessing a broader range of downstream tasks. Incorporating analyses of zero-shot classification, VQA2.0, and GQA tasks would offer a more comprehensive perspective.**
>
> Our score and debiasing method are designed to calculate the similarity score between an image and a caption. Thus, it does not naturally support VQA tasks.
>
> While BLIP was neither designed nor benchmarked on zero-shot classification, we follow your suggestion to evaluate our methods on ImageNet1K and show that our $\alpha$-debiasing solution (using a fixed $\alpha=1$) can nearly double the performance from 18.6% to 36.2%. We also try to search for alpha by sampling an additional one-shot validation set (and repeat using 3 random seeds). This extremely low-shot validation set can lead us to a near-optimal alpha, achieving 40.0% without retraining or finetuning the model:
>
>
> | Method                                  | Result      |
> |-----------------------------------------|-------------|
> | ITCScore                                | 31.7       |
> | ITMScore                                | 37.4       |
> | VisualGPTScore ($\alpha=0$)             | 18.6       |
> | VisualGPTScore ($\alpha=1$)             | 36.2       |
> | VisualGPTScore ($\alpha^*_{val}=0.65\pm0.01$) | 40.0$\pm$0.1  |
> | VisualGPTScore ($\alpha^*_{test}=0.69$)  | 40.2       |
>
> In this experiment, by sampling only a single Gaussian noise image for the calculation of $P_{train}(t)$, we incur a negligible inference cost, equivalent to the testing of one additional image.
>
> > **Question 4: Is it plausible that negative captions rarely occur in web corpora? This factor might be affecting the performance of "blind models" and deserves further investigation.**
>
> You are right that negative captions in benchmarks such as ARO rarely occur in web corpora. This explains why even “blind models” are able to outperform SOTA methods on many recent popular benchmarks. As said by Reviewer ANt2, our experiments suggest that *“existing benchmarks are in some sense, still not “hard enough” and contain correlations that can be exploited or can be solved by using language priors”*. We hope our experiments are useful for everyone designing new benchmarks for vision-language tasks.

---

> > ### Comment · Reviewer_AoET · 2023-11-22
> > **Response to rebuttal**
> >
> > Thank you for the detailed response. This clarifies several doubts and misunderstandings. Also, appreciate the additional experiments on the zero-shot classification task.
> >
> > Overall I like the insights from this paper. If we fix the value of $\alpha^*$, then still results are still consistent and better.
> > But my understanding of the train-test bias is still different.
> >
> > For weakness 1, my understanding is that we need to measure the $\alpha$ only when the test domain is different than i.i.d. So, what kind of distribution shifts are we observing across these datasets? Some examples of such differences might give the proper motivation.
> > If I consider the Fig.1 as a motivation example then trying to solve the train-test distribution shift needs to be further justified. Scenario 2 focuses on the issue of "language prior" and calculating $\alpha^*$ does not justify whether this bias is removed or not.
> >
> > Is it possible to provide some before/after probability distribution examples -- success & failure cases?
> >
> >
> > Note: Increased the score to 5. I will decide the final score based on the needed clarification on the above question.

---

> > > ### Author Response · Authors · 2023-11-22
> > >
> > > Thank you for your positive feedback and appreciating our insights!
> > >
> > > > **For weakness 1, my understanding is that we need to measure the $\alpha$ only when the test domain is different than i.i.d. So, what kind of distribution shifts are we observing across these datasets? Some examples of such differences might give the proper motivation.**
> > >
> > > In this paper, we focus on the train-test marginal shift in $P(text)$. This type of distribution shift is very common in well-constructed vision-language benchmarks that enforce a balanced $P_{test}(text)$. Here is a simple example: In ImageNet1K, $P_{test}(text)$ is the same for all 1000 classes, as each class has the same number of testing images. However, VLM’s training data  (image-text pairs found on the Web) tend to be naturally imbalanced. For instance, $P_{train}(cat)$ is much higher than $P_{train}(frog)$ because people tend to upload more cat images than frog images to the Web. As such, to account for the fact that $P_{test}(cat) = P_{test}(frog)$ in balanced test sets such as ImageNet, we propose dividing our score by $P_{train}(text)^\alpha$ (e.g., $\alpha$=1) to improve performance. We see this strategy is effective for other balanced VLM benchmarks such as Winoground and EqBen (as shown in Table 3).
> > >
> > > However, for popular testing benchmarks such as ARO, negative captions are constructed to be implausible and thus rare in VLM's training data too. For example, the negative caption $t_{negative}$ of  “white a duck spreads its wings in while the water” is equally unlikely to occur in Web data (i.e., $P_{train}(t_{negative}) \approx 0 = P_{test}(t_{negative})$). Therefore, dividing by $P_{train}(text)^\alpha$ (where $\alpha$>0) hurts performance because it increases the match score for negative captions.
> > >
> > > Moving forward, we believe that balanced benchmarks such as Winoground and EqBen may be better testbeds for compositional reasoning (as evidenced by the lower SOTA performance), and so conjecture that debiasing (with a val-tuned $\alpha$ or even by naively assuming $\alpha$ = 1) will continue to be useful.
> > >
> > > Please let us know if these answers clarify your questions. We sincerely appreciate your engagement!

---

### Official Review · Reviewer_ANt2 · 2023-10-30

**Soundness:** 4 excellent
**Presentation:** 4 excellent
**Contribution:** 4 excellent
**Rating:** 8
**Confidence:** 5

**Summary:**

This is a scientific work that empirically analyzes language bias in image-text retrieval tasks and generative vision-language models (not image generation models, but text generation models). They first characterize the ability of generative vision-language models to match images to text in a zero-shot manner by measuring the probability that a textual sequence may be generated from an image. They then turn to a benchmark-centric view, empirically showing that several benchmarks can be solved even by blind LLMs in this manner, simply by their ability to flag linguistically unlikely captions from language priors. They show that with postprocessing, generative approaches can outperform handcrafted discriminative approaches on image-text retrieval tasks, even highly compositional ones.

**Strengths:**

It is an open question to what degree vision-language models are doing the task of image-text retrieval as opposed to exploiting spurious correlations. This is dependent on the degree to which benchmarks themselves can be beaten by exploiting correlations. The major strength of this paper is that they (at least partially) answer this question. The experiments are convincing and cover a broad range of vision-language models. They show that even blind LLMs and VLMs do surprisingly well on these benchmarks, suggesting that existing benchmarks are in some sense, still not "hard enough" and contain correlations that can be exploited or can be solved by using language priors.

The proposed method for debiasing vision-language models works well, given the simplicity. I consider the simplicity and generality a strength.

The scientific conclusions of this paper are novel and useful for everyone designing new benchmarks for vision-language tasks.

**Weaknesses:**

A minor weakness of the paper is that they do not compare with the recent crop of truly "large" generative vision-language models like BLIP-2, LLAVA, etc. However, this is a minor weakness and I do not think it needs to be really addressed in this work, since these models are still new enough that training them is extra engineering work. Also, if anything, the language prior should be worse in LLM-based VLMs.

**Questions:**

I have no questions.

---

> ### Author Response · Authors · 2023-11-14
> **Thank you!**
>
> We sincerely appreciate Reviewer ANt2's positive and insightful review. Your understanding and articulation of our work's core contributions are highly encouraging. We agree that exploring the extent to which benchmarks can be solved by exploiting language priors is a critical question, and we are glad that our work contributes meaningfully to this discussion. We are also grateful for your recognition of the simplicity and generality of our debiasing approach.
>
> We address your (minor) concern below:
> > **A minor weakness of the paper is that they do not compare with the recent crop of truly "large" generative vision-language models like BLIP-2, LLAVA, etc. However, this is a minor weakness and I do not think it needs to be really addressed in this work, since these models are still new enough that training them is extra engineering work. Also, if anything, the language prior should be worse in LLM-based VLMs.**
>
> We have included the results of the state-of-the-art captioning model BLIP-2 (both stage-1-QFormer and stage-2-FlanT5) in Appendix D. For example, Table 9 shows that our debiasing solution, even with a fixed alpha = 1, consistently improves performance on balanced benchmarks across all model variants. For your convenience, we attach the image-to-text retrieval results (text score) for Winoground and EqBen below:
>
> | Winoground                          | $\alpha=0$ | $\alpha=1$ | $\alpha=\alpha^*$ |
> | ------------------------------ | ---------- | ---------- | ----------------- |
> | BLIP-1                         | 27.0       | 33.0       | 36.5              |
> | BLIP-2 (stage-1-QFormer)       | 24.3       | 29.3       | 33.0              |
> | BLIP-2 (stage-2-FlanT5)        | 25.3       | 31.5       | 34.3              |
>
> | EqBen                          | $\alpha=0$ | $\alpha=1$ | $\alpha=\alpha^*$ |
> | ------------------------------ | ---------- | ---------- | ----------------- |
> | BLIP-1                         | 9.6       | 19.8       | 19.8              |
> | BLIP-2 (stage-1-QFormer)       | 12.2       | 21.9       | 22.2              |
> | BLIP-2 (stage-2-FlanT5)        | 8.5       | 22.0       | 22.0              |

---

### Official Review · Reviewer_85bY · 2023-10-31

**Soundness:** 2 fair
**Presentation:** 3 good
**Contribution:** 1 poor
**Rating:** 5
**Confidence:** 4

**Summary:**

The paper proposes a simple method to use the class of generative VLM for image to text similarity tasks. Although the proposed method is quite general, the paper evaluates it on the BLIP model and, as previously observed in the literature, shows that some Vision-Language benchmarks can be solved by only looking at the text modality. The authors also propose to reduce the reliance of VLMs on the language prior by using a probabilist post-processing calibration technique aimed at controlling the amount of linguistic bias of generative VLMs which is shown to improve image-to-text search results on the proposed benchmarks.

**Strengths:**

The idea of calibrating a pre-trained generative VLM models to adapt to the “format” of the test dataset is interesting and impactful.

**Weaknesses:**

1. **(Lack of novelty)**: Measuring similarity using the average log likelihood of strings given a fixed context (an image+prompt in this case) is not new. For example in the language community it has been previously used multiple times to provide alternative similarity measures to using encoder models (based on dot product similarity). Furthermore, such idea has already been extended to VLMs in [1], where it has been shown that “models trained on captioning can perform on-par with models trained with the usual contrastive image-text matching loss.” What is particularly novel about "VisualGPTScore"?
2. **(Soundeness)** The paper briefly points out a connection between the proposed calibration approach and Mutual Information based approaches. How does this connection help the reader? What is the intuition that motivates using point-wise mutual information to improve the calibration of the deployed generative VLMs? Can the authors comment more on this? As of now, this seems more an afterthought rather than a clear and sound motivation.


**Minor:**

While the proposed VisualGPTScore is more efficient to be computed than next-token generation it is fair to point out that it is much slower than computing similarity scores based on dot products (e.g. CLIP) especially when the size of the retrieval index grows. See for example, [1, 2] and their techniques to limit the computational cost of performing image-to-text search over large databases. Can the authors comment more on this in the manuscript?

[1] Antonie Miech et al. “Thinking Fast and Slow: Efficient Text-to-Visual Retrieval with Transformers”

[2] J. Li et al., “Align before fuse: Vision and language representation learning with momentum distillation”

**Questions:**

1. The proposed trick to estimate the marginal over text p(t) is not sound. Why should averaging Gaussian noise fed as input to the VLM work more efficiently than averaging over the distribution of natural images? Is there any theoretical guarantee that this is the correct thing to do? Especially given the fact that Gaussian noise has never been used during training of the VLM and is therefore out of distribution for the model.
    - I suggest the authors to perform the experiments using a more recent VLM like (LLaVA or BLIPv2, also BLIP is not SoTA) which both can be directly used to compute the language marginals.
2. If VLMs are deployed as Zero Shot models why do we care about the gap between test and training (p_te vs p_tr) since the model do not use p_tr?
3. I am not fully convinced by the empirical evaluation. Isn’t it obvious that the proposed method with the optimal alpha performs better than any other method since it has been optimized (through Cross Validation) to find the best possible alpha for each dataset independently? I’d expect a comparison with other baseline methods that calibrate the model’s predictions before inference.

---

> ### Author Response · Authors · 2023-11-14
> **Rebuttal (1/2)**
>
> After reading through the comments, we believe Reviewer 85bY’s concerns may stem from some misunderstandings about our work. We first clarify:
>
> > **Question 2: If VLMs are deployed as zero-shot models, why do we care about the gap between test and training (p_te vs p_tr) since the model does not use p_tr?**
>
> As we focus on zero-shot applications of generative VLMs, $P_{train}$ in our paper refers to **VLMs’ pretraining datasets**, such as LAION for BLIPv1 and BLIPv2. In fact, most of the recent popular benchmarks such as ARO and SugarCrepe are designed to evaluate VLMs in a zero-shot fashion, and thus do not even provide a training split.
>
> > **Question 1.1: I suggest the authors perform the experiments using a more recent VLM like (LLaVA or BLIPv2, also BLIP is not SoTA) which both can be directly used to compute the language marginals.**
>
> The language marginal ($P_{train}(t)$) refers to **VLMs’ pretraining captions**. Therefore, without sampling any images, one simply cannot approximate this marginal from the language decoder of LLaVA or BLIPv2, whose language models are pretrained on text corpora different from VLMs’ pretraining captions.
>
> Also, our paper already includes SOTA captioning-model BLIPv2 results (both stage-1-Q-Former and stage-2-FlanT5-XL). For instance, in Table 9, we show that our $\alpha$-debiasing solution consistently improves all BLIP model variants even with a fixed $\alpha$=1.
>
> > **(Lack of novelty) Because this generative score has been applied before, what is particularly novel about our work (e.g., compared to paper [1])?**
>
> We agree that the generative score has been applied in prior art such as [1]. In fact, our goal is **not** to introduce a new method. Instead, we aim to **revisit** the language priors and biases in generative VLMs (as noted by Reviewer ANt2). This issue has been **neglected** in recent popular image-text retrieval benchmarks until now.
>
> We also carefully review the paper [1] and note two crucial differences:
>
>
> - While [1] focuses on training this score from scratch, we focus more on its zero-shot application, especially on how to improve its performance under a substantial distribution (marginal) shift between VLMs’ pretraining data and downstream tasks.
> - [1] only applies this generative score to text-to-image retrieval tasks (as we also show in Table 3-b). However, it chooses **not** to report image-to-text retrieval performance, although it uses the same COCO and Flickr30K benchmarks. We think this omission is likely due to the issue of language biases, which our paper thoroughly addresses. Concretely, we also provide algorithms for debiasing the generative score from [1], which dramatically improves performance on challenging benchmarks (such as COCO and Flickr30K).
>
> > **(Soundness) Is our calibration approach motivated by pointwise mutual information (PMI)? How does the connection between our approach and PMI help the reader?**
>
> To clarify, our approach (dividing by the marginal $P_{train}(t)^\alpha$) is motivated by our analysis in Section 3 (not by PMI). Here is a concrete example (cf. Figure 1) for ease of understanding: in some recent compositionality benchmarks such as SugarCrepe, a negative caption such as “*people are cooking in a kitchen*” is more common in the VLM’s trainset (thus a higher $P_{train}(t)$) than the positive caption “*people are posing in a kitchen*”. As such, dividing by $P_{train}(t)$ forces the model to select the caption that matches better to the image (rather than the caption with a larger marginal prior).
>
> Our Appendix A shows that our approach $\frac{P(t|i)}{P(t)^\alpha}$ happens to be the same as smoothed estimates of pointwise mutual information (PMI$^k$), which also reduces the effect of marginal priors from training data. We include this connection because some readers may find it interesting.
>
> > **(Minor: Efficiency) VisualGPTScore is slower than CLIPScore especially when the size of the retrieval index grows. [1,2] show some methods for reducing this cost. Can the authors comment more on this in the manuscript?**
>
> While efficiency is not the focus of our work, we are happy to cite [1,2] that suggest re-ranking or distillation to reduce the inference cost. This could be a promising future direction.
>
> > **Question 1: The proposed trick to estimate the marginal over text p(t) is not sound. Why should averaging Gaussian noise fed as input to the VLM work more efficiently than averaging over the distribution of natural images? Is there any theoretical guarantee that this is the correct thing to do? Especially given the fact that Gaussian noise has never been used during training of the VLM and is therefore out of distribution for the model.**
>
> Our empirical trick is motivated by [3] that approximates P(answer) using a null prompt “N/A”. Yet, the language models used in [3] also are never trained on P(answer | “N/A”). Studying its theoretical guarantee could be an interesting future work.

---

> > ### Author Response · Authors · 2023-11-14
> > **Rebuttal (2/2)**
> >
> > > **Question 3: I am not fully convinced by the empirical evaluation. Isn’t it obvious that the proposed method with the optimal alpha performs better than any other method since it has been optimized (through Cross Validation) to find the best possible alpha for each dataset independently? I’d expect a comparison with other baseline methods that calibrate the model’s predictions before inference.**
> >
> >  In some sense, one can view our approach as a method for calibrating the influence of the language prior. We are not aware of other baselines for calibration - e.g., Platt scaling or naively adding a tune-able temperature parameter to the softmax predictions of the language model would not change the image-to-text retrievals.
> >
> > Finally, we would also like to point out that one can use a fixed alpha = 1 without cross validation to achieve a consistent and reasonable performance increase for all (balanced) benchmarks.
> >
> >
> > **References**:
> >
> > [1] Antonie Miech et al. “Thinking Fast and Slow: Efficient Text-to-Visual Retrieval with Transformers”.
> >
> > [2] J. Li et al., “Align before fuse: Vision and language representation learning with momentum distillation”
> >
> > [3] Zhao et al. “Calibrate Before Use: Improving Few-Shot Performance of Language Models”

---

### Official Review · Reviewer_eKtF · 2023-11-01

**Soundness:** 3 good
**Presentation:** 3 good
**Contribution:** 2 fair
**Rating:** 5
**Confidence:** 3

**Summary:**

This paper investigates the zero-shot performance of generative VLMs in image-text retrieval tasks. A novel metric, VisualGPTScore, is introduced to compute the match score by generating a specific text string based on an image. Notably, the authors identify the train-test distribution shift and present a probabilistic post-processing method. This approach enables the regulation of linguistic bias in generative VLMs during testing without necessitating model retraining or fine-tuning. The proposed method sets new state-of-the-art results on several image-to-text retrieval benchmarks.

**Strengths:**

1. Applying generative VLMs to image-to-text retrieval tasks is an intriguing endeavor.
2. Comprehensive experiments have been conducted, achieving state-of-the-art results.
3. The problem's formulation as a train-test distribution shift, followed by a probabilistic derivation, leading to the adjustable parameter alpha, is both logical and intriguing.

**Weaknesses:**

1. My primary concern is the application's real-world viability. Image-text retrieval, often utilized in search engines, demands high time efficiency. With this method, for every new image, the VLM must process all texts, resulting in substantial computational costs. In contrast, traditional methods like CLIP pre-compute text embeddings and only require a dot product with each image embedding. Hence, while the experimental results are impressive, I question this method's practical value.
2. As demonstrated in Table 7 in the appendix, as the dataset size increases, the OTS scores progressively deteriorate, and the gap with ITM widens even when using the optimal alpha. How can this be explained? Might this indicate an inherent limitation of the method?

**Questions:**

1. Following the first weakness, could you provide a time-efficiency assessment comparing various methods?
2. The paper mentions, "To apply this to our generative VLMs, we choose to sample 'null' inputs as Gaussian noise images." Why are Gaussian noise images suitable as "null" inputs?

---

> ### Author Response · Authors · 2023-11-14
> **Rebuttal**
>
> We appreciate Reviewer eKtF’s feedback. We will address your concerns below:
>
> > **My primary concern is the application's real-world viability. Image-text retrieval, often utilized in search engines, demands high time efficiency. With this method, for every new image, the VLM must process all texts, resulting in substantial computational costs. In contrast, traditional methods like CLIP pre-compute text embeddings and only require a dot product with each image embedding. Hence, while the experimental results are impressive, I question this method's practical value.**
>
> While our proposed score is slow for *large-scale* retrieval, it can still be practical for other applications like evaluating text-to-image generation, e.g., measuring the similarity score between a text prompt and a generated image. As an illustration, our method can complement the widely used CLIPScore [1,2], which struggles with understanding complicated texts that involve compositions of objects, attributes, and their relations. In fact, recent image-text retrieval benchmarks, such as ARO and SugarCrepe, are designed specifically to evaluate VLMs’ compositional reasoning capabilities.
>
> Since our primary focus is on studying the issues of linguistic priors in both generative VLMs and vision-language benchmarks, we leave it to future work to adapt our method for more efficient large-scale retrieval tasks. For instance, one could use established re-ranking technique or distill the VisualGPTScore into CLIPScore to improve inference speed [3].
>
> > **Following the first weakness, could you provide a time-efficiency assessment comparing various methods?**
>
> We provide a time-efficiency comparison between ITCScore (CLIPScore), ITMScore, and VisualGPTScore, using the same BLIP model for large-scale inference (1000 images x 5000 texts):
>
> | Metric          | Inference Cost     | Time on Flickr30K |
> |-----------------|--------------------|----------------------------------------------------|
> | ITCScore        | O(\|image\| + \|text\|) | 2 minutes                                         |
> | ITMScore        | O(\|image\| x \|text\|) | 3 hours                                           |
> | VisualGPTScore  | O(\|image\| x \|text\|) | 3 hours                                           |
>
>
> > **As demonstrated in Table 7 in the appendix, as the dataset size increases, the OTS scores progressively deteriorate, and the gap with ITM widens even when using the optimal alpha. How can this be explained? Might this indicate an inherent limitation of the method?**
>
> Good question! We include Table 7 (retrieval performance on training set) to motivate a scientific discussion on the inherent linguistic priors of VisualGPTScore, which make it a biased estimator of $P_{train}(image | text)$. For a thorough discussion, please refer to Appendix C, titled *“Is VisualGPTScore a biased estimator?”*. Intuitively, such estimators can suffer from well-known “long-tailed” issues that arise when learning from imbalanced datasets (i.e., in our case, certain sentences are more common than others). We hope that our initial discussion on this issue can serve as a useful reference point for future work.
>
> > **The paper mentions, "To apply this to our generative VLMs, we choose to sample 'null' inputs as Gaussian noise images." Why are Gaussian noise images suitable as "null" inputs?**
>
> Our empirical findings echoes [4], which uses a language model to approximate P(text) = P(text | “N/A”). Their null text prompt “N/A” is also not seen during training, but can empirically help calibrate language models. Also, we try both gaussian noise images and training set images, and find that they show similar performance (Table 4).
>
>
> **References**:
>
> [1] Ruiz et al. “DreamBooth: Fine Tuning Text-to-Image Diffusion Models for Subject-Driven Generation”.
>
> [2] Brooks et al. “InstructPix2Pix: Learning to Follow Image Editing Instructions”.
>
> [3] Antonie Miech et al. “Thinking Fast and Slow: Efficient Text-to-Visual Retrieval with Transformers”.
>
> [4] Zhao et al. “Calibrate Before Use: Improving Few-Shot Performance of Language Models”

---

> > ### Comment · Reviewer_eKtF · 2023-11-20
> >
> > Thanks for your reply!
> >
> > 1. As for the "null" inputs, you mentioned that "Their null text prompt “N/A” is also not seen during training". However, in natural language corpora, N/A is likely to appear, and I didn't find any relevant expressions in [4] (please correct me if I overlooked it). Additionally, Table 3 in [4] demonstrates that the choice of content-free input does affect accuracy. Therefore, conducting more in-depth analyses and experiments on this is highly recommended.
> >
> > 2. You mentioned that this method is practical for other applications, such as complementing the widely used CLIPScore". I suggest conducting experiments on more tasks to prove that the method has performance improvements, as only text-image retrieval tasks are evaluated in the paper.

---

> > > ### Author Response · Authors · 2023-11-21
> > > **Thank you for the feedback**
> > >
> > > We thank Reviewer eKtF for your follow-up comments. We address your remaining concerns below:
> > >
> > > > **As for the "null" inputs, you mentioned that "Their null text prompt “N/A” is also not seen during training". However, in natural language corpora, N/A is likely to appear, and I didn't find any relevant expressions in [4] (please correct me if I overlooked it). Additionally, Table 3 in [4] demonstrates that the choice of content-free input does affect accuracy. Therefore, conducting more in-depth analyses and experiments on this is highly recommended.**
> > >
> > > To clarify, we provide some concrete examples from [4] for the sentence classification task, which used null input strings such as “N/A”, “abc”, or even “dasjhasjkdhjskdhds” to approximate the marginal:
> > >
> > > - P(answer) $\approx$ P(answer | “N/A was born in”)
> > > - P(answer) $\approx$ P(answer | “abc was born in”)
> > > - P(answer) $\approx$ P(answer | “dasjhasjkdhjskdhds was born in”)
> > >
> > > While these null text strings themselves might be included in the pretraining corpora, they are certainly out-of-distribution for calculating the above marginals.
> > >
> > > We appreciate Reviewer eKtF’s suggestion to explore other content-free inputs. We also conduct tests using **mean training image** (in place of gaussian noise images) and find they yield almost identical performance. Concretely, we sample random sets of 100 or 1000 images from the trainset and calculate their average (which visually resembles gaussian noise images too). The table below shows that both **mean training images** and gaussian noise images yield similar performance on Winoground (following Table 4’s setup):
> > >
> > >
> > > | Method               | Gaussian Noise | Mean of 100 train images | Mean of 1000 train images |
> > > |----------------------|-----------|------------|-------------|
> > > | $\alpha=\alpha_{val}^*$  |    36.4$\pm$0.1       |      36.2$\pm$0.4       |      36.3$\pm$0.3        |
> > >
> > >
> > > > **You mentioned that this method is practical for other applications, such as complementing the widely used CLIPScore". I suggest conducting experiments on more tasks to prove that the method has performance improvements, as only text-image retrieval tasks are evaluated in the paper.**
> > >
> > > We want to point out that recent compositionality benchmarks like Winoground are specifically designed for image-text evaluation within the context of image-text retrieval. In these benchmarks, human annotators assign a score of 1 to an (image, text) pair if it matches, or 0 if it does not. This means that **a higher performance on these benchmarks corresponds to a better alignment with human judgements**. To demonstrate the effectiveness of our proposed method, we present the Winoground results for two recent text-to-image evaluation algorithms, LLMScore [5] and VPEval [6], both of which claim to complement CLIPScore. Our results demonstrate that our proposed score exhibits stronger compositional understanding than these heavily engineered methods that use expensive ChatGPT during inference.
> > >
> > > | Method                     | Text Score | Image Score | Group Score |
> > > |----------------------------|------------|-------------|-------------|
> > > | CLIPScore                  | 28.5       | 10.8        | 12.8        |
> > > | VisualGPTScore |  36.5       | 21.5        | 16.8        |
> > > | LLMScore (GPT4 + BLIPv2)     | 21.3       | 17.8        | 12.5        |
> > > | VPEval (GPT3.5 + BLIPv2)     | 12.8       | 11.0        | 6.3         |
> > >
> > > **References**:
> > >
> > > [5] Lu et al. LLMScore: Unveiling the Power of Large Language Models in Text-to-Image Synthesis Evaluation. 2023.
> > >
> > > [6] Cho et al. Visual Programming for Text-to-Image Generation and Evaluation. 2023.